# TRUST-REGION ADAPTIVE POLICY OPTIMIZATION

**Mingyu Su**[1], **Jian Guan**[2], **Yuxian Gu**[1], **Minlie Huang**[1], **Hongning Wang**[1]*
[1]Conversational AI (CoAI) Group, Tsinghua University [2]Ant Group
sumy25@mails.tsinghua.edu.cn, hw-ai@tsinghua.edu.cn

## ABSTRACT

Post-training methods, especially Supervised Fine-Tuning (SFT) and Reinforcement Learning (RL), play an important role in improving large language models' (LLMs) complex reasoning abilities. However, the dominant two-stage pipeline (SFT then RL) suffers from a key inconsistency: SFT enforces rigid imitation that suppresses exploration and induces forgetting, limiting RL's potential for improvements. We address this inefficiency with TRAPO (**T**rust-**R**egion **A**daptive **P**olicy **O**ptimization), a hybrid framework that interleaves SFT and RL within each training instance by optimizing SFT loss on expert prefixes and RL loss on the model's own completions, unifying external supervision and self-exploration. To stabilize training, we introduce Trust-Region SFT (TrSFT), which minimizes forward KL divergence inside a trust region but attenuates optimization outside, effectively shifting toward reverse KL and yielding stable, mode-seeking updates favorable for RL. An adaptive prefix-selection mechanism further allocates expert guidance based on measured utility. Experiments on five mathematical reasoning benchmarks show that TRAPO consistently surpasses standard SFT, RL, and SFT-then-RL pipelines, as well as recent state-of-the-art approaches, establishing a strong new paradigm for reasoning-enhanced LLMs. Our code and data are publicly available at https://github.com/Su-my/TRAPO.

## 1 INTRODUCTION

Driven largely by post-training, especially Supervised Fine-Tuning (SFT) and Reinforcement Learning (RL) techniques, large language models (LLMs) have demonstrated substantial advances in complex reasoning tasks, such as mathematical deduction (Lightman et al., 2023), program synthesis (Chen et al., 2021), and multi-step decision-making (Shridhar et al., 2021), shifting them from surface-level chat-bots toward deep reasoners (OpenAI, 2025; Guo et al., 2025). Particularly, current dominant LLM post-training pipeline integrates SFT and RL in a two-stage fashion, where SFT precedes RL. This design counts on the SFT stage to teach an LLM imitating carefully curated expert demonstrations, and then sharpens the model's reasoning skills via trial-and-error in the RL stage.

However, a fundamental barrier exists that hinders the synergy between SFT and RL. On the one hand, SFT tends to lock the trained models into imitative and rigid behavior modes (Chu et al., 2025; Chen et al., 2025), which hinders effective exploration crucially needed in the RL stage. On the other hand, SFT is also prone to cause catastrophic forgetting in the trained models, impeding the RL stage from exploiting the pretraining knowledge for improvements. These inconsistencies raise an important challenge: *how can we effectively incorporate the knowledge-distillation benefits of SFT into RL training without undermining the model's exploratory capacity and pretrained knowledge?*

In this work, we pinpoint the root cause of the aforementioned barrier to be the disjoint two-stage SFT and RL training, and propose a hybrid post-training framework as our solution. Specifically, standard SFT training requires "sufficient" reduction of its training loss, but this is not sufficiently an indicative metric for later RL training. In other words, a lower SFT loss does not suggest a better starting point for RL training. Worse still, over-training in SFT might push the model out of the region suited for RL, while there is no immediate signal measuring that. Our idea is then to interleave SFT with RL in each training instance: only perform SFT on the prefix of the given expert

---

*Corresponding Author

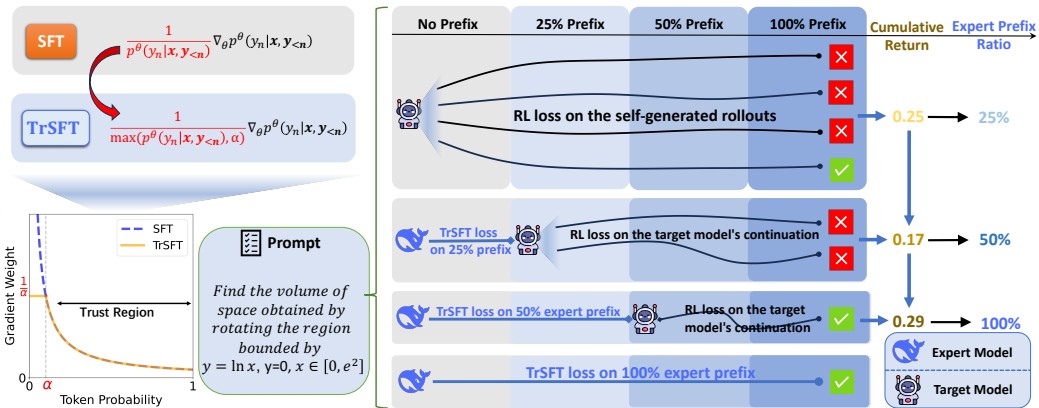

Figure 1: An overview of our **TRAPO** framework, which synergistically combines two key components: Trust-Region SFT (TrSFT) and Adaptive Expert Guidance. **Left:** By clipping the gradient weight with a trust-region parameter $\alpha$, TrSFT prevents exploding gradients on low-probability tokens, ensuring a stable learning signal when combining with RL. **Right:** The adaptive guidance mechanism implements a "learn-while-practicing" loop. When the target model fails a rollout, its cumulative return dynamically dictates the length of an expert prefix provided for guidance. The model then continues generation and the full trajectory is optimized using both the TrSFT loss on expert prefixes and a standard RL objective on the model rollout.

trajectory, and continue RL training by completing the trajectory thereafter. In this way, we can best integrate the utility of expert demonstrations and self-exploration at the finest grain.

Though intuitive, several technical challenges prevent a straightforward combining scheme. First, using the language of RL, the negative log-likelihood (NLL) objective in SFT aims to minimize the forward Kullback–Leibler (KL) divergence between the target and expert policies (i.e., behavior cloning (Torabi et al., 2018)), which exhibits a strong mode-covering property by assigning relatively high probabilities even to regions where the expert policy has no support (Minka et al., 2005; Malinin & Gales, 2019). This is especially toxic when one interleaves SFT with RL in a per-instance basis, as those inflated modes immediately cause degenerations in the target policy (e.g., repetition or erroneous decoding) and keeps RL away from effective exploration. In this work, to mitigate the negative impact of SFT on RL, we propose a new form of SFT, named Trust-Region SFT (TrSFT), which minimizes forward KL divergence inside a trust region, while outside the region it downweights the optimization strength and effectively shifts the objective toward reverse KL. As reverse KL minimization is characterized by mode-seeking behaviors (Gu et al., 2023), TrSFT avoids blindly balancing the modes in the entire space, and only seeks the most prominent mode of the expert policy when the expert policy is clearly distinct from the current target policy. This provides a stable and robust policy for the following RL training.

Second, determining the optimal amount of expert guidance for each training instance is nontrivial. A "one-size-fits-all" prefix length is inherently inefficient: it stifles valuable exploration on problems the model could solve independently, while offering insufficient support on challenging problems where more guidance is crucial. To address this, we introduce an adaptive prefix-selection mechanism that dynamically allocates expert guidance based on its utility. The core idea is a scaffolding approach: the model first attempts a problem without any guidance. If it struggles, indicated by a low return, we incrementally provide longer expert prefixes in subsequent rollouts for that same problem. This strategy ensures that expert guidance is used judiciously only as much as needed, thereby creating a dynamic balance between self-exploration and expert guidance.

Building on these designs, we propose **TRAPO** (**T**rust-**R**egion **A**daptive **P**olicy **O**ptimization), which unifies SFT and RL in a principled way. Extensive experiments on five mathematical reasoning benchmarks demonstrate the substantial benefits of our approach. TRAPO delivers significant gains over conventional post-training algorithms. Specifically, it outperforms standalone SFT and pure RL by +6.3 and +6.2 points, respectively. More critically, TRAPO surpasses the highly competitive SFT-then-RL baseline by +2.3 points, proving the superiority of our integrated framework over the disjoint two-stage approach. Crucially, we show that a naive, direct combination of

SFT and RL objectives is counterproductive, causing a catastrophic performance collapse of over 18 points compared to the pure RL baseline. This result validates our initial hypothesis and highlights the necessity of our TrSFT objective for achieving a stable and synergistic integration. Furthermore, our pass@k analysis reveals that TRAPO, unlike pure RL methods, effectively expands the model's underlying solution space, leading to stronger test-time scaling properties. These compelling results establish TRAPO as a new, effective paradigm for developing reasoning-enhanced LLMs.

## 2 METHODOLOGY

Our proposed framework, TRAPO, guides RL by leveraging prefixes from offline expert trajectories (Figure 2). These prefixes serve a dual purpose: they act as in-context demonstrations to guide exploration and as direct supervision signals for skill internalization. The successful implementation of TRAPO hinges on resolving two key challenges: **(1) Guidance internalization:** how to effectively learn from the prefixes, and **(2) Guidance selection:** how to choose the optimal prefix length for each prompt. We will now provide an overview of the framework before detailing each component.

### 2.1 OVERVIEW OF TRAPO

The design of TRAPO is to realize a synergistic "learn-while-practicing" paradigm for LLM post-training, which couples learning from offline expert trajectories with online RL updates. The core workflow unfolds as follows: for each prompt, (1) a selected prefix from an offline expert trajectory is provided as a starting context; (2) the target policy rolls out from there to complete the reasoning; and (3) a dual update is then performed, where the generated completion is used for standard RL updates, while the expert prefix is used for direct policy optimization to internalize the expert's reasoning skills. TRAPO is expected to guide the target policy to find high return completions via the selected expert prefixes (Liu et al., 2025a;

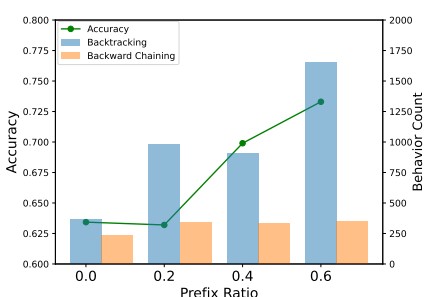

Figure 2: Accuracy and characteristic of Qwen2.5-3B-Instruct reasoning on MATH-500 with different amount of tokens from DeepSeek-R1 as prefixes.

Huang et al., 2025) and pruning unproductive solution space via skill internalization from the expert. As shown in Figure 2, we empirically validate the benefit of introducing prefixes from expert trajectories in target policy rollouts. On the MATH-500 benchmark, we provided Qwen2.5-3B-Instruct with prefixes from DeepSeek-R1, and then calculated the response accuracy and counted the frequency of two key reasoning behaviors, i.e., backtracking and backward chaining (Gandhi et al., 2025), in the completed suffixes. Detailed experimental procedures are provided in Appendix C.1. It is clearly observed that longer expert prefixes steadily improve accuracy and stimulate the emergence of advanced reasoning behaviors.

However, integrating RL with expert demonstration presents two significant challenges: **(1) Effective learning objective for guidance internalization.** The aim of incorporating expert prefixes is to not only immediately guide the target policy to find high return trajectories, but also enable the target policy to distill the expert's core problem-solving skills, highlighting the need for an effective joint learning algorithm. **(2) Efficient and adaptive guidance selection.** When using expert prefixes to guide exploration, a "one-size-fits-all" strategy, which applies a uniform prefix selection and a fixed number of rollouts to all prompts (Liu et al., 2025a; Huang et al., 2025), can hardly be optimal. This simple method would over-guide in easy prompts, thus stifling valuable exploration, while under-guiding in difficult ones, leading to failed rollouts. It remains challenging to design an adaptive, prompt specific mechanism that allocates only the minimal guidance necessary to ensure the target policy can effectively discover the correct solution paths.

### 2.2 TRUST-REGION SFT

Regarding the first challenge of internalizing an expert's reasoning skills, a straightforward approach is to combine the standard SFT loss on expert prefixes with an RL objective on model-generated rollouts. However, this naive combination leads to severe performance degradation in our experiments (see §3.3). To uncover the root cause, we first conduct a pilot study of SFT's training dynamics.

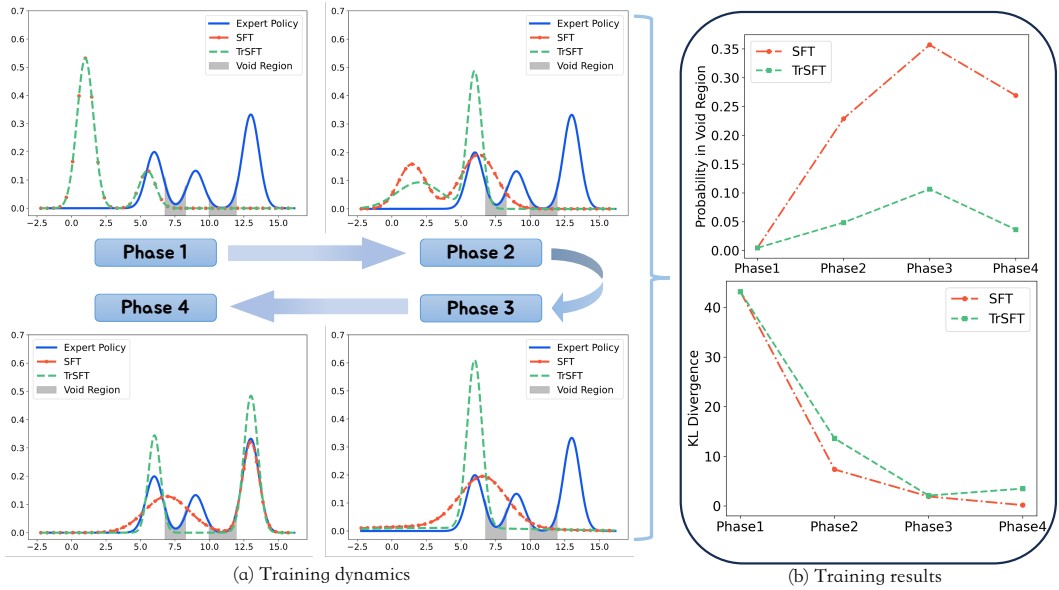

(a) Training dynamics          (b) Training results

Figure 3: An illustrative experiment showing the training dynamics during SFT. Panel (a) shows snapshots of learnt target policy at four consecutive training phases, corresponding to training steps of 0, 50, 100, and 1000, respectively. Panel (b) presents the KL divergence curve along with the change in cumulative probability of the target policy within the void regions.

The findings motivate our proposed Trust-Region SFT (TrSFT), an objective designed to resolve this instability, whose theoretical properties we then analyze.

**Training Dynamics of SFT**    We begin by examining the standard SFT training objective. Given an expert policy $p_{\mathrm{E}}$ and target policy $p_{\mathrm{T}}^{\theta}$ parameterized by $\theta$, SFT forces $p_{\mathrm{T}}^{\theta}$ to mimic $p_{\mathrm{E}}$ by minimizing the NLL in every trajectory $\boldsymbol{y} = (y_1, \cdots, y_n)$ from the expert policy in a given prompt set $X$,

$$L_{\mathrm{SFT}}(\theta) = \mathbb{E}_{\boldsymbol{x} \sim X} \left[ \mathbb{E}_{\boldsymbol{y} \sim p_{\mathrm{E}}(\cdot | \boldsymbol{x})} \left[ -\log p_{\mathrm{T}}^{\theta}(\boldsymbol{y} | \boldsymbol{x}) \right] \right]. \tag{1}$$

The SFT objective is equivalent to minimizing the cumulative token-level forward KL divergence (see Appendix A.1 for derivation). To understand its training dynamics, we conduct an illustrative experiment training a two-mode Gaussian Mixture Model (GMM) to mimic a three-mode expert GMM. As shown in Figure 3, this process reveals the distribution-blending phenomenon (Minka et al., 2005; Malinin & Gales, 2019): the target policy assigns probability to void regions unsupported by either policy (e.g., shaded regions in Figure 3 (a)). This effect is detrimental to RL, as these regions lead to degenerated outputs (e.g., repetitions) that hinder effective exploration.

We point out that the underlying cause of this phenomenon lies in the gradient of standard SFT loss:

$$\nabla_\theta L_{\mathrm{SFT}} = -\frac{1}{N} \sum_{i=1}^{N} \sum_{n=1}^{|\boldsymbol{y}^i|} \frac{1}{p_{\mathrm{T}}^{\theta}(y_n^i | \boldsymbol{x}^i, \boldsymbol{y}_{<n}^i)} \nabla_\theta p_{\mathrm{T}}^{\theta}(y_n^i | \boldsymbol{x}^i, \boldsymbol{y}_{<n}^i). \tag{2}$$

The term $\frac{1}{p_{\mathrm{T}}^{\theta}(y_n^i | \boldsymbol{x}^i, \boldsymbol{y}_{<n}^i)}$ in Eq (2) can be considered as the weight of training token $y_n^i$ given context $(\boldsymbol{x}^i, \boldsymbol{y}_{<n}^i)$. When $y_n^i$ is drawn from a mode of expert policy which is far away from the current policy's mode, e.g., the rightmost mode of expert policy in Figure 3, this weight term will be extremely large. Consequently, the resulting $\nabla_\theta L_{\mathrm{SFT}}$ is more detrimental than helpful for $p_{\mathrm{T}}^{\theta}$ update, though it points to the correct the direction of policy improvement: the inflated gradient tends to first push $p_{\mathrm{T}}^{\theta}$ into the voided regions during the online gradient update of SFT, as shown in Figure 3.

While sufficient training can eventually correct for distribution-blending in standard SFT (e.g., phase 4 in Figure 3), this issue is critical in TRAPO. Here, the interleaving of SFT and RL means that any probability mass allocated to void regions will immediately cause degenerated rollouts. This can significantly hinder model learning, making modifications to the standard SFT objective essential.

**Clipping the gradient of standard SFT loss**  To mitigate the negative impact of distribution-blending during online model update, we propose Trust-Region SFT (TrSFT), a mechanism designed to adaptively leverage the SFT updates. The core idea is to establish a region where the gradient from standard SFT loss can be trusted, while intervention is needed outside this region to prevent model collapsing into undesirable parts of the solution space:

$$\nabla_\theta L_{\text{TrSFT}}^\alpha = -\frac{1}{N} \sum_{i=1}^{N} \sum_{n=1}^{|\boldsymbol{y^i}|} \frac{1}{\max\left(p_{\text{T}}^\theta(y_n^i|\boldsymbol{x^i}, \boldsymbol{y_{<n}^i}), \alpha\right)} \nabla_\theta p_{\text{T}}^\theta(y_n^i|\boldsymbol{x^i}, \boldsymbol{y_{<n}^i}), \tag{3}$$

where $\alpha \in [0, 1]$ is a hyperparameter defining the boundary of trust region. The optimization objective exhibits several desirable properties:

**(1) Safe knowledge instillation within the trust region.** Standard SFT risks distribution blending by applying large updates towards "distant" modes of the expert policy. TrSFT mitigates this by defining a dynamic trust region based on the target policy's own beliefs (i.e., whether $p_{\text{T}}^\theta(y_n^i|\boldsymbol{x^i}, \boldsymbol{y_{<n}^i}) \geq \alpha$). Within this region, it employs the standard SFT objective to aggressively mimic expert policy's behavior. Outside the region, the constant weight $\frac{1}{\alpha}$ significantly dampens the gradients, thereby reducing the disruptive impact of large gradient updates on the target policy's immediate behaviors.

The benefit of TrSFT can be easily understood by our illustrative experiment in Figure 3. As illustrated in Phases 2 and 3, TrSFT prioritizes learning from high-overlapping regions—modes jointly favored by $p_{\text{E}}$ and $p_{\text{T}}^\theta$. This helps the target policy first instills expert's wisdom that is "close" and readily tangible, while preserving its existing strengths. This conservative, trust-region-based approach is safer and more stable than aggressive, full-distribution matching, especially when the target policy's capacity is limited.

**(2) Leading to a reasonable optimization endpoint.** We present the optimization problem defined by the gradient defined in Eq. (3), and theoretically derive the solution.

**Proposition 1.** *Let $S(\lambda) = \{ c \mid p_E(c) > \alpha\lambda, c \in \mathcal{C} \}$ and $\mathcal{C}$ is the vocabulary. There exists a unique $\lambda \in (0, 1)$ such that $\lambda = \sum_{c \in S(\lambda)} p_E(c)$ . And for this $\lambda$, the optimal solution of the optimization problem defined by the gradient defined in Eq.* (3) *is given by*

$$p_T^*(c) = \begin{cases} \dfrac{p_E(c)}{\lambda}, & \text{if } p_E(c) > \alpha\lambda. \\ 0, & \text{otherwise} \end{cases}$$

The optimal solution for TrSFT counters distribution-blending by pruning low-probability regions in expert policy ($p_{\text{T}}^*(c) = 0$) and rescaling primary modes ($p_{\text{T}}^*(c) = p_{\text{E}}(c)/\lambda$). This dual action effectively transforms the objective from mode-covering of Forward KL to mode-seeking akin to reverse KL, forcing the policy to focus on the expert's core skills and thereby facilitating high-return rollouts for RL. Appendix A.2 shows more details.

## 2.3 MICRO-GROUP SAMPLING

To elegantly address the second challenge of guidance selection, we propose *micro-group sampling*, which adaptively allocates guidance from expert prefixes based on the observed returns from the current policy rollouts, thereby both minimizing unnecessary reliance on expert prefixes and accommodating the heterogeneity of prompt difficulty within each training batch.

As illustrated in Figure 1, in each training prompt, TRAPO creates $N$ micro-groups in order, where each micro-group $g_i$ (for $i = 1, \ldots, N$) is specified by three key hyper-parameters: the prefix length ratio $L_i$, the return threshold $t_i$, and the sampling budget $n_i$. For micro-group $g_i$, TRAPO first computes the average return from all samples generated in the preceding micro-groups. If the average return is smaller than the threshold $t_i$, TRAPO provides the current target policy with a prefix whose length is set by the ratio $L_i$ of the complete expert trajectory, and then samples $n_i$ completions from the target policy. Otherwise, no expert prefix is provided, and $n_i$ policy rollouts are obtained directly from the target policy.

We set $0 = L_1 < L_2 < \cdots < L_N = 1$. $L_1 = 0$ is to ensure that in each training prompt, TRAPO always starts from guidance-free self-exploration RL, and $L_N = 1$ allows the target policy, when necessary, to access the complete reasoning path from the expert. As a result, an increasing level of $L_i$ ensures richer guidance is provided only when shorter prefixes prove insufficient. For clarity, Appendix B details the complete training procedure specified by TRAPO in Algorithm 1.

## 3 EXPERIMENTS

### 3.1 EXPERIMENTAL SETUP

**Training Details**   Our primary training dataset is OpenR1-Math-46k-8192 (Yan et al., 2025), which consists of a large collection of verified reasoning trajectories generated by DeepSeek-R1 for complex mathematical problems. To enhance the diversity of guidance, we additionally pair each problem with another trajectory sampled from OpenR1-Math-200k (Face, 2025). Following recent work (Yan et al., 2025; Huang et al., 2025; Fu et al., 2025), we use Qwen2.5-Math-7B (Yang et al., 2024b) as the base model. To further validate the generality of our approach, we additionally evaluate it on Qwen2.5-7B-Instruct (Yang et al., 2024a), a general-purpose model.

**Implementation Details**   We adopt the Group Relative Policy Optimization (GRPO) (Shao et al., 2024; Liu et al., 2025c) algorithm without the KL penalty (Hu et al., 2025) for RL. The training is configured with a batch size of 128 and a constant learning rate of $5 \times 10^{-6}$. Our adaptive guidance mechanism operates on a total group size of 8 following prior work, which is partition into four micro-groups of sizes $\{4, 2, 1, 1\}$. These micro-groups correspond to relative expert prefix-length proportions of $(L_1, \ldots, L_4) = (0, 0.2, 0.5, 1.0)$ and are activated by reward thresholds of $(t_1, \ldots, t_4) = (-1, 0.5, 0.7, 0.9)$, respectively. The threshold $t_1 = -1$ ensures the first micro-group is always without guidance. For our TrSFT objective, the trust-region parameter $\alpha$ in Eq. (3) is set to 0.1. More details can be found in Appendix C.2.

**Evaluation benchmark and metrics**   We focus on mathematical reasoning tasks, while also evaluating various methods on both mathematical and general-domain benchmarks. Specifically, the mathematical benchmarks include AIME2024 (Li et al., 2024), AMC (He et al., 2024), Minerva (Lewkowycz et al., 2022), OlympiadBench (He et al., 2024), and MATH-500 (Hendrycks et al., 2021). Given the relatively small number of test samples in AIME2024 and AMC, we report avg@32 on these benchmarks, while employing pass@1 for the remaining three. For general-domain reasoning benchmarks, we report pass@1 on ARC-c (Clark et al., 2018) and MMLU-Pro (Wang et al., 2024) to examine whether the improvements in reasoning ability generalize to other reasoning tasks.

**Baselines**   We consider two categories of baseline methods:

- **Pure RL without External Expert Guidance.** This category includes GRPO (Shao et al., 2024), PRIME-Zero (Cui et al., 2025), SimpleRL-Zero (Zeng et al., 2025), OpenReasoner-Zero (Hu et al., 2025), and Oat-Zero (Liu et al., 2025c).
- **RL Incorporating External Expert Guidance.** This category includes (1) SFT: directly training the model to imitate expert trajectories, (2) SFT-then-RL: following the standard two-stage pipeline where SFT precedes RL, (3) LUFFY (Yan et al., 2025): augmenting each group of eight rollouts with one expert trajectory to perform offline RL, and (4) ReLIFT (Ma et al., 2025): alternating between SFT and RL training across different batches, thereby dynamically switching the optimization objective.

### 3.2 MAIN RESULTS

**Mathematical reasoning performance**   As shown in Table 1, TRAPO achieves an average score of **56.6** across five mathematical reasoning benchmarks, outperforming all baselines. In particular, TRAPO yields improvements of **+6.3** and **+6.2** over *SFT* and *GRPO*, respectively, and a **+2.3** gain over the *SFT-then-RL* baseline. These results validate our core hypothesis: TRAPO effectively enables the model to both internalize expert skills and leverage guidance for superior exploration, leading to a more robust acquisition of reasoning abilities.

Table 1: Main experiment results on mathematical and general reasoning benchmarks based on **Qwen2.5-Math-7B**. **Bold** and underline indicate the best and second-best results, respectively. * means the results are taken from the corresponding paper.

| Model | Mathematical Reasoning | | | | | | General Domain Reasoning | | |
|---|---|---|---|---|---|---|---|---|---|
| | AIME2024 | AMC | MATH-500 | Minerva | Olympiad | Avg. | ARC-c | MMLU-Pro | Avg. |
| Qwen2.5-Math-7B | 11.9 | 33.7 | 47.0 | 12.5 | 21.9 | 25.4 | 34.5 | 23.3 | 28.9 |
| Qwen2.5-Math-7B-Instruct | 11.3 | 48.2 | 82.6 | 36.8 | 39.7 | 43.7 | 72.4 | 38.3 | 55.4 |
| *Pure RL without External Expert Guidance* | | | | | | | | | |
| GRPO | 24.0 | 59.0 | 84.0 | 39.3 | 45.8 | 50.4 | 80.5 | 47.2 | 63.9 |
| PRIME-Zero* | 17.0 | 54.0 | 81.4 | 39.0 | 40.3 | 46.3 | 73.3 | 32.7 | 53.0 |
| SimpleRL-Zero* | 27.0 | 54.9 | 76.0 | 25.0 | 34.7 | 43.5 | 30.2 | 34.5 | 32.4 |
| OpenReasoner-Zero* | 16.5 | 52.1 | 82.4 | 33.1 | 47.1 | 46.2 | 66.2 | **58.7** | 62.5 |
| Oat-Zero* | 33.4 | 61.2 | 78.0 | 34.6 | 43.4 | 50.1 | 70.1 | 41.7 | 55.9 |
| *RL Incorporating External Expert Guidance* | | | | | | | | | |
| SFT | 27.7 | 56.0 | 84.8 | 38.2 | 44.7 | 50.3 | 51.5 | 33.1 | 42.3 |
| SFT-then-RL | **33.5** | 62.3 | 86.6 | 41.2 | 47.6 | 54.3 | 52.9 | 36.1 | 44.5 |
| ReLIFT | 28.2 | 64.9 | 87.4 | 33.8 | 52.5 | 53.4 | 76.2 | 52.5 | 64.4 |
| LUFFY | 29.4 | 65.5 | 88.4 | 38.2 | 56.0 | 55.5 | 80.5 | 53.0 | 66.7 |
| **Our Method** | | | | | | | | | |
| TRAPO | 28.3 | **66.2** | **89.2** | **41.5** | **57.6** | **56.6** | **83.7** | 52.8 | **68.3** |

Figure 4: Training dynamics of TRAPO compared with GRPO. From left to right: average reward, generation length, and output entropy during training. For fair comparison, both reward and generation length are computed by excluding trajectories guided by expert prefixes.

**General-domain reasoning performance** On the two general reasoning benchmarks, TRAPO attains an average score of **68.3**, surpassing all baselines. By contrast, *SFT* and *SFT-then-RL* exhibit notably lower scores on these benchmarks, indicating that TRAPO, while leveraging external guidance, does not confine the model to rigid reasoning patterns; instead, it yields stronger generalization.

**Training dynamics** Figure 4 presents a comparative analysis of the training dynamics between TRAPO and GRPO, revealing three key advantages of our approach: (1) TRAPO consistently achieves higher rewards throughout the training process and finally converges to a significantly higher final reward level. (2) The generation length curve highlights that TRAPO rapidly increases its output length in the early stages, indicating a swift internalization of the expert's extended reasoning patterns. In contrast, GRPO struggles to produce longer solutions, consistently maintaining

Table 2: Ablation study on TRAPO components.

| Model | AIME2024 | AMC | MATH-500 | Minerva | Olympiad | Avg. |
|---|---|---|---|---|---|---|
| GRPO (Qwen2.5-Math-7B) | 24.0 | 59.0 | 84.0 | 39.3 | 45.8 | 50.4 |
| + Micro-group sampling | 26.0 | 63.7 | 84.2 | 39.3 | 50.1 | 52.7 |
| + Micro-group sampling + *SFT Loss* | 14.6 | 32.8 | 60.2 | 25.4 | 28.5 | 32.3 |
| + Micro-group sampling + *LUFFY Loss* | 26.2 | 65.1 | 87.2 | 36.8 | 52.5 | 53.6 |
| + Micro-group sampling + *TrSFT Loss* | 28.3 | 66.2 | 89.2 | 41.5 | 57.6 | 56.6 |

a short output length. (3) While both methods show an initial drop in policy entropy, their long-term behavior differs. TRAPO stabilizes at a relatively higher entropy level. We attribute this to its ability to maintain a dynamic balance: it simultaneously refines its own high-probability reasoning paths while remaining open to learning from externally provided, potentially low-probability expert guidance.

**Extending to general-purpose LLMs.** To further validate the generality of TRAPO, we conduct additional experiments on Qwen2.5-7B-Instruct, a general-purpose instruction-tuned model. As shown in Figure 5, TRAPO still achieves higher average accuracy than all other baselines across five mathematical reasoning benchmarks.

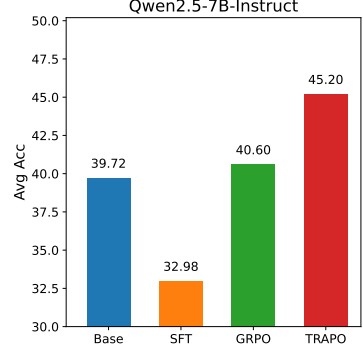

### 3.3 ABLATION STUDY

To assess the contributions of micro-group sampling and TrSFT, we conduct an ablation study on Qwen2.5-Math-7B (Table 2). Results show that micro-group sampling alone, even without explicit prefix learning, surpasses GRPO by adaptively determining prefix length to enhance reasoning and reward density. Adding TrSFT further improves performance: unlike standard SFT loss (which degrades) or the offline RL loss from LUFFY (which brings limited gains), TrSFT effectively internalizes expert prefixes. We also analyze the effect of the trust-region parameter $\alpha$ in Appendix D.

Figure 5: Average accuracy across five mathematical benchmarks.

### 3.4 TEST-TIME SCALING

We evaluate pass@k, the success rate over k independent rollouts, to better estimate the upper bound of model capability (Snell et al., 2024), as recent studies show that multiple generation attempts reveal reasoning potential more accurately than few rollouts (Yue et al., 2025a; Wang et al., 2022).

Figure 6 illustrates the pass@k performance on the AIME2024 benchmark, from which we derive two key insights: (1) We observe that the base model (Qwen2.5-Math-7B) surpasses the GRPO-trained model when evaluated with a sufficiently large $k$. This aligns with prior findings (Yue et al., 2025a) and suggests that standard RL primarily stimulates the model to select better solutions from its existing knowledge space, but does not funda-

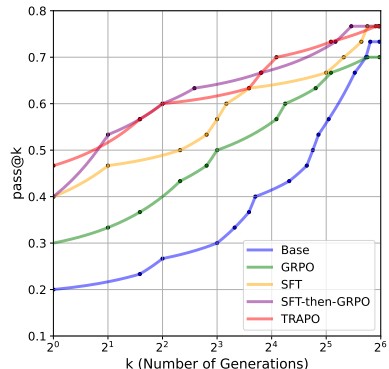

Figure 6: Comparison of pass@k results on the AIME2024 dataset

mentally expand that space with new problem-solving skills. (2) Both TRAPO and the SFT-based methods demonstrate strong performance scaling with larger $k$, indicating they possess a richer underlying solution space. The superior performance of TRAPO highlights its success in effectively internalizing the external knowledge from expert trajectories, thereby expanding the model's intrinsic capabilities.

## 4 RELATED WORKS

**RL for LLM reasoning**  RL has recently driven major advances in LLM reasoning, as demonstrated by milestone systems such as OpenAI-o1 (Jaech et al., 2024), DeepSeek-R1 (Guo et al., 2025), and Kimi-1.5 (Team et al., 2025). Follow-up studies have examined RL from three complementary perspectives. (1) Empirical analyses investigate its effect on reasoning, e.g., Yue et al. (2025a) shows that RL refines existing solution trajectories rather than expanding reasoning capabilities. (2) Data-centric approaches reshape training signals, such as R3 (Xi et al., 2024) with reverse curriculum learning, and ADARFT (Shi et al., 2025) or Logic-R1 (Xie et al., 2025) with dynamic difficulty scheduling to improve sample efficiency. (3) Optimization methods refine RL objectives, from PPO's clipped trust region and GRPO's group-based advantage estimation to recent variants like DAPO (Yu et al., 2025), Dr.GRPO (Liu et al., 2025c), and VAPO (Yue et al., 2025b). Distinct from these directions, TRAPO enhances RL efficiency by adaptively incorporating expert demonstrations, which both guide exploration and act as direct supervision, simultaneously improving reward density and reasoning capability.

**Combining SFT with RL**  While the traditional training paradigm primarily followed a serial approach of first applying SFT and then RL, recent works have shifted focus towards concurrently performing both SFT and RL. A straightforward approach is to directly combine the SFT and RL losses, adjusting their respective weights. For instance, SRFT (Fu et al., 2025) proposes a single-stage training method that dynamically adjusts the weights of SFT and RL losses based on token entropy. AMFT (He et al., 2025) views the balance between SFT and RL as a learnable parameter and introduces a meta-gradient adaptive weight controller. HPT (Lv et al., 2025), on the other hand, dynamically decides whether to apply SFT or RL based on the model's rollout performance, with binary weights of 0 or 1. Another approach involves interleaving SFT within the RL pipeline, as seen in ReLIFT (Ma et al., 2025), which collects poorly performing samples during RL training and stores them in a buffer to later apply SFT. Additionally, LUFFY (Yan et al., 2025), inspired by offline RL, treats one expert trajectory as offline data and mixes it with the remaining seven online trajectories in a group, using an importance ratio to calibrate the distribution shift, achieving state-of-the-art performance. Similar to our work, Prefix-RFT (Huang et al., 2025) samples prefixes as guidance and uses entropy to select expert tokens for SFT. These works collectively emphasize the significant benefits of combining these two post-training paradigms. We present the first study to theoretically investigate the challenge of combining SFT and RL objectives, and propose a simple but effective solution to resolve the distribution-blending effect and foster synergy with the RL objective.

## 5 CONCLUSION

In this work, we propose TRAPO, a one-stage training paradigm that unifies SFT and RL under expert prefix guidance. TRAPO enables both the internalization of guidance and its use in generating high-quality continuations. To stabilize training, we introduce TrSFT, which constrains policy updates within a trust region, mitigating the disruptive effect of low-probability tokens and effectively shifting Forward KL's mode-covering behavior toward Reverse KL's mode-seeking behavior. TrSFT integrates seamlessly into RL to enhance guidance utilization. We further design micro-group sampling, which adaptively adjusts guidance length based on return improvements, balancing exploration with expert supervision. Experiments on mathematical reasoning tasks show that TRAPO significantly outperforms standalone SFT, RL, and the conventional SFT-then-RL pipeline.

## 6 REPRODUCIBILITY STATEMENT

To ensure the reproducibility of our experimental results, we have open-sourced our training and evaluation code, including detailed training scripts and configurations, at: `https://github.com/Su-my/TRAPO`. The specific training and evaluation settings are described in §3.1 and Appendix C.2.

ACKNOWLEDGMENT

This work was supported by the National Science Foundation of China (under Key Project No. 92570203), Beijing Natural Science Foundation (under project No. Z250001) and the National Key Research and Development Program of China (under project No. 2024YFC3606800).

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

# A DERIVATIONS AND PROOFS

## A.1 DERIVATION OF THE EQUIVALENCE BETWEEN SFT AND KL

Following Agarwal et al. (2024), we remove the length normalization factor and define the discrepancy between token-level distribution of $p_E$ and $p_T$ over a given sequence $\boldsymbol{y}$:

$$\mathcal{D}\big(p_E\|p_T^\theta\big)(\boldsymbol{y}|\boldsymbol{x}) := \sum_{n=1}^{|\boldsymbol{y}|} \mathcal{D}\big(p_E(\cdot|y_{<n},\boldsymbol{x})\|p_T^\theta(\cdot|y_{<n},\boldsymbol{x})\big). \tag{4}$$

The problem then becomes proving the following equivalence:

$$\min_\theta L_{\text{SFT}}(\theta) \Leftrightarrow \min_\theta \mathbb{E}_{x\sim X}\Big[\mathbb{E}_{\boldsymbol{y}\sim p_E(\cdot|\boldsymbol{x})}\big[\mathcal{D}_{\text{KL}}\big(p_E\|p_T^\theta\big)(\boldsymbol{y}|\boldsymbol{x})\big]\Big]. \tag{5}$$

We start by substituting Eq. (4) into the objective expression on the right-hand side of Eq. (5):

$$\mathbb{E}_{\boldsymbol{y}\sim p_E(\cdot|\boldsymbol{x})}\big[\mathcal{D}_{\text{KL}}\big(p_E\|p_T^\theta\big)(\boldsymbol{y}|\boldsymbol{x})\big] = \mathbb{E}_{\boldsymbol{y}\sim p_E(\cdot|\boldsymbol{x})}\big[\sum_{n=1}^{|\boldsymbol{y}|} \mathcal{D}_{\text{KL}}\big(p_E(\cdot|y_{<n},\boldsymbol{x})\|p_T^\theta(\cdot|y_{<n},\boldsymbol{x}))\big] \tag{6}$$

$$= \mathbb{E}_{\boldsymbol{y}\sim p_E(\cdot|\boldsymbol{x})}\big[\sum_{n=1}^{|\boldsymbol{y}|}\sum_{y_n\in V} p_E(y_n|y_{<n},\boldsymbol{x})\log\frac{p_E(y_n|y_{<n},\boldsymbol{x})}{p_T^\theta(y_n|y_{<n},\boldsymbol{x})}\big]. \tag{7}$$

We then remove the term that are independent of $\theta$:

$$\min_\theta \mathbb{E}_{\boldsymbol{x}\sim X}\Big[\mathbb{E}_{\boldsymbol{y}\sim p_E(\cdot|\boldsymbol{x})}\big[\sum_{n=1}^{|\boldsymbol{y}|}\sum_{y_n\in V} p_E(y_n|y_{<n},\boldsymbol{x})\log\frac{p_E(y_n|y_{<n},\boldsymbol{x})}{p_T^\theta(y_n|y_{<n},\boldsymbol{x})}\big]\Big] \tag{8}$$

$$\Leftrightarrow \min_\theta \mathbb{E}_{\boldsymbol{x}\sim X}\Big[\mathbb{E}_{\boldsymbol{y}\sim p_E(\cdot|\boldsymbol{x})}\big[\sum_{n=1}^{|\boldsymbol{y}|}\sum_{y_n\in V} -p_E(y_n|y_{<n},\boldsymbol{x})\log p_T^\theta(y_n|y_{<n},\boldsymbol{x})\big]\Big] \tag{9}$$

$$\Leftrightarrow \min_\theta \mathbb{E}_{\boldsymbol{x}\sim X}\Big[\mathbb{E}_{\boldsymbol{y}\sim p_E(\cdot|\boldsymbol{x})}\big[\sum_{n=1}^{|\boldsymbol{y}|}\mathbb{E}_{y_n\sim p_E(\cdot|y_{<n},\boldsymbol{x})}[-\log p_T^\theta(y_n|y_{<n},\boldsymbol{x})]\big]\Big] \tag{10}$$

$$\Leftrightarrow \min_\theta \mathbb{E}_{\boldsymbol{x}\sim X}\Big[\mathbb{E}_{\boldsymbol{y}\sim p_E(\cdot|\boldsymbol{x})}\big[\sum_{n=1}^{|\boldsymbol{y}|}[-\log p_T^\theta(y_n|y_{<n},\boldsymbol{x})]\big]\Big] \tag{11}$$

$$\Leftrightarrow \min_\theta \mathbb{E}_{\boldsymbol{x}\sim X}\Big[\mathbb{E}_{\boldsymbol{y}\sim p_E(\cdot|\boldsymbol{x})}\big[-\log p_T^\theta(\boldsymbol{y}|\boldsymbol{x})\big]\Big] \tag{12}$$

The last line follows from the equivalence between the logarithm of a product and the sum of logarithms.

## A.2 PROOF OF PROPOSITION 1

**Proposition 1.** *Let $S(\lambda) = \{c \mid p_E(c) > \alpha\lambda, c \in \mathcal{C}\}$ and $\mathcal{C}$ is the vocabulary. There exists a unique $\lambda \in (0,1)$ such that $\lambda = \sum_{c\in S(\lambda)} p_E(c)$ . And for this $\lambda$, the optimal solution of the optimization problem defined by the gradient defined in Eq. (3) is given by*

$$p_T^*(c) = \begin{cases} \dfrac{p_E(c)}{\lambda}, & \text{if } p_E(c) > \alpha\lambda. \\ 0, & \text{otherwise} \end{cases}$$

*Proof.* First, we present the equivalent form of this optimization problem, with the derivation similar to that in Eq. (5):

$$\min_\theta L_{\text{TrSFT}}^\alpha(\theta) \Leftrightarrow \min_\theta \mathbb{E}_{x\sim X}\Big[\mathbb{E}_{\boldsymbol{y}\sim p_E(\cdot|\boldsymbol{x})}\big[\mathcal{D}_{\text{TrSFT}}^\alpha\big(p_E\|p_T^\theta\big)(\boldsymbol{y}|\boldsymbol{x})\big]\Big],$$

$$\text{where} \quad \mathcal{D}_{\text{TrSFT}}^\alpha(p_E\|p_T^\theta) = \sum_{c\in\mathcal{C}} p_E(c)\ell^\alpha\big(p_T^\theta(c)\big) \quad \text{and} \quad \ell^\alpha(p) = \begin{cases} -\dfrac{p}{\alpha} + 1 - \log\alpha, & p < \alpha, \\ -\log p, & p \geq \alpha, \end{cases}$$

Then we apply the Karush–Kuhn–Tucker (KKT) (Ghojogh et al., 2021) conditions.

We introduce multiplier $\lambda \in R$ for the equality constraint and multipliers $\{\mu_c | c \in \mathcal{C}\}$ for nonnegativity constraints. The Lagrangian is

$$\mathcal{L}\big(p_{\mathrm{T}}(c_1), \cdots, p_{\mathrm{T}}(c_N), \lambda, \mu_{c_1}, \cdots, \mu_{c_N}\big) = \sum_c p_{\mathrm{E}}(c)\ell^\alpha\big(p_{\mathrm{T}}(c)\big) - \sum_c \mu_c p_{\mathrm{T}}(c) + \lambda\Big(\sum_c p_{\mathrm{T}}(c) - 1\Big).$$

(13)

For each $c$, stationarity gives $p_{\mathrm{E}}(c)\nabla_{p_{\mathrm{T}}(c)}\ell^\alpha\big(p_{\mathrm{T}}(c)\big) - \mu_c + \lambda = 0$, together with $\mu_c \geq 0$ and $\mu_c p_{\mathrm{T}}^c = 0$. The derivative of $\ell^\alpha$ is

$$\nabla_{p_{\mathrm{T}}}\ell^\alpha(p_{\mathrm{T}}) = \begin{cases} -\dfrac{1}{\alpha}, & 0 \leq p_{\mathrm{T}} \leq \alpha, \\ -\dfrac{1}{p_{\mathrm{T}}}, & p_{\mathrm{T}} > \alpha. \end{cases}$$

**Case A $\big(p_{\mathbf{T}}(c) > \alpha\big)$.**

In this case, $\nabla\ell^\alpha\big(p_{\mathrm{T}}(c)\big) = -1/p_{\mathrm{T}}(c)$. The stationarity condition with $\mu_c = 0$ gives

$$-\frac{p_{\mathrm{E}}(c)}{p_{\mathrm{T}}(c)} + \lambda = 0 \quad \Rightarrow \quad p_{\mathrm{T}}(c) = \frac{p_{\mathrm{E}}(c)}{\lambda}.$$

Since we assumed $p_{\mathrm{T}}(c) > \alpha$, this requires $p_{\mathrm{E}}(c) > \alpha\lambda$.

**Case B $\big(0 < p_{\mathbf{T}}(c) \leq \alpha\big)$.**

Here $\nabla\ell^\alpha(p_{\mathrm{T}}(c)) = -1/\alpha$. With $\mu_c = 0$, the condition becomes

$$-\frac{p_{\mathrm{E}}(c)}{\alpha} + \lambda = 0 \quad \Rightarrow \quad p_{\mathrm{E}}(c) = \alpha\lambda.$$

Thus interior solutions in $(0, \alpha]$ are only possible if $p_{\mathrm{E}}(c) = \alpha\lambda$.

**Case C $\big(p_{\mathbf{T}}(c) = 0\big)$.**

At the boundary $p_{\mathrm{T}}(c) = 0$, we use $\nabla\ell^\alpha(0^+) = -1/\alpha$. Stationarity then gives

$$-\frac{p_{\mathrm{E}}(c)}{\alpha} + \lambda - \mu_c = 0 \quad \Rightarrow \quad \mu_c = \lambda - \frac{p_{\mathrm{E}}(c)}{\alpha}.$$

For feasibility we require $\mu_c \geq 0$, hence this case occurs only when $p_{\mathrm{E}}(c) \leq \alpha\lambda$.

Collecting the above conditions, the optimal solution must satisfy:

$$p_{\mathrm{T}}^\star(c) = \begin{cases} \dfrac{p_{\mathrm{E}}(c)}{\lambda}, & \text{if } p_{\mathrm{E}}(c) > \alpha\lambda, \\ 0, & \text{if } p_{\mathrm{E}}(c) < \alpha\lambda, \\ \text{any value in } [0, a], & \text{if } p_{\mathrm{E}}(c) = \alpha\lambda. \end{cases}$$

In realistic token distributions the set of tokens satisfying $p_{\mathrm{E}}(c) = \alpha\lambda$ has measure zero and can be ignored. Equivalently, one may simply enforce $p_{\mathrm{T}}(c) = 0$ for all such tokens. The optimality conditions derived above remain valid. Then from $\sum_c p_{\mathrm{T}}(c) = 1$, we can derive $\lambda = \sum_{c \in S(\lambda)} p_{\mathrm{E}}(c)$.

Observe that the normalization condition requires

$$\sum_{c \in S(\lambda)} \frac{p_{\mathrm{E}}(c)}{\lambda} = 1.$$

As $\lambda$ increases continuously from 0 to 1, the left-hand side decreases monotonically from $+\infty$ to the sum of probablity less than 1, while the right-hand side remains constant at 1. Therefore, by the intermediate value theorem, such a value of $\lambda$ must exist.

$\square$

---

**Algorithm 1** TRAPO : Micro-group Sampling with Adaptive Guidance

---

**Require:** Dataset $D_{\text{data}} = \{(\boldsymbol{x}, \boldsymbol{y}^\star)\}$; expert policy $p_E$; target policy $p_T^\theta$; trust-region parameter $\alpha$;
    number of micro-groups $N$; metadata $\{(n_i, L_i, t_i)\}_{i=1}^N$ with $0 = L_1 < L_2 < \cdots < L_N = 1$
**Ensure:** Updated student parameters $\theta$
 1: **function** PASSRATE($\mathcal{S}$)
 2:     **return** $\frac{1}{\max(1,|\mathcal{S}|)} \sum_{(\boldsymbol{x},\boldsymbol{y}) \in \mathcal{S}} \mathbb{1}[\text{Correct}(\boldsymbol{y}; \boldsymbol{y}^\star)]$          $\triangleright$ compare to ground truth $\boldsymbol{y}^\star$
 3: **end function**
 4: **for** each mini-batch $\mathcal{B} \subset D_{\text{data}}$ **do**
 5:     **for** each prompt $\boldsymbol{x} \in \mathcal{B}$ **do**
 6:         $\mathcal{S} \leftarrow \emptyset$                          $\triangleright$ collected samples for this $x$ so far
 7:         **for** $i = 1$ to $N$ **do**                       $\triangleright$ serial micro-groups
 8:             $pr_i \leftarrow$ PASSRATE($\mathcal{S}$)     $\triangleright$ use samples from previous $1, \ldots, i{-}1$ micro-groups
 9:             **if** $pr_i \leq t_i$ **then**                    $\triangleright$ inject expert guidance
10:                 $\boldsymbol{y} \leftarrow p_E(\cdot \mid \boldsymbol{x})$
11:                 $\boldsymbol{y}^{<L_i} \leftarrow$ first $\lfloor L_i \cdot |\boldsymbol{y}| \rfloor$ tokens of $\boldsymbol{y}$
12:                 **for** $k = 1$ to $n_i$ **do**
13:                     $\hat{\boldsymbol{y}} \sim p_T^\theta(\cdot \mid \boldsymbol{x}, \boldsymbol{y}^{<L_i})$
14:                     $\mathcal{S} \leftarrow \mathcal{S} \cup \{(\boldsymbol{x}, \boldsymbol{y}^{<L_i} \oplus \hat{\boldsymbol{y}})\}$
15:                 **end for**
16:             **else**                            $\triangleright$ no guidance
17:                 **for** $k = 1$ to $n_i$ **do**
18:                     $\hat{\boldsymbol{y}} \sim p_T^\theta(\cdot \mid \boldsymbol{x})$
19:                     $\mathcal{S} \leftarrow \mathcal{S} \cup \{(\boldsymbol{x}, \hat{\boldsymbol{y}})\}$
20:                 **end for**
21:             **end if**
22:         **end for**
23:     **end for**
24:     Compute $\nabla_\theta L_{\text{TrSFT}}^\alpha$ on *guided tokens only* (those fed to the target) via Eq. (3)
25:     Compute $\nabla_\theta L_{\text{GRPO}}$ on *self-generated trajectories*
26:     $\nabla_\theta L \leftarrow \nabla_\theta L_{\text{TrSFT}}^\alpha + \nabla_\theta L_{\text{GRPO}}$
27:     $\theta \leftarrow \text{OptimizerStep}(\theta, \nabla_\theta L)$
28: **end for**

---

# B   PSEUDOCODE FOR TRAPO

Algorithm 1 summarizes the complete training procedure of TRAPO.

# C   EXPERIMENTAL DATAILS

## C.1   PRELIMINARY EXPERIMENT

**Data Collection**   We use DeepSeek-R1 to generate eight solutions for each problem in the MATH-500 benchmark. For each solution, we extract prefixes with varying token proportions as guidance, which are then concatenated to the original prompt. At each token proportion level, we obtain 4,000 prompts, which are subsequently fed into Qwen2.5-3B-Instruct for continuation.

**Evaluation**

- Accuracy is computed by comparing the generated answers with the ground truth using Math-Verify .

- We further input the generated continuations into GPT-4o-mini to count the occurrences of two distinct reasoning behaviors. The specific prompt templates used are as follows:

---

**Backtracking**

You are given a math reasoning problem and the latter half of a reasoning trace generated by Qwen2.5-3B-Instruct.
**Problem:** {problem_text}
**Model output:** {output_text}
Detect BACKTRACKING: moments where the model abandons a line of attack and starts a new, distinct approach (not merely the next algebraic step of the same plan). Count an instance when the text signals a restart/switch such as:

- "That doesn't work / leads nowhere / contradiction, so try...", "Instead, I'll...", "Another approach:", "Restart with...", "Consider a different method/case".
- Dropping the current construction and trying a fresh one (new substitution, inequality, identity, factorization, case split that resets the plan).

Do NOT count routine sequential steps in one coherent plan as backtracking. Return only the number between `<count>` and `</count>`. If none, return `<count>0</count>`.

---

**Backward Chaining**

You are given a math reasoning problem and the latter half of a reasoning trace generated by Qwen2.5-3B-Instruct.
**Problem:** {problem_text}
**Model output:** {output_text}
Detect BACKWARD-CHAINING: reasoning that starts from (or is framed by) the desired conclusion/target and derives necessary conditions that would make it true. Count an instance when the text says things like:

- "We want to show P, so it suffices to show Q", "Work backwards from the target...", "If the result were true, then ... must hold", "To get X, we need Y", rearranging *from the goal form to requirements*.
- Proof-by-contradiction setup qualifies if it explicitly frames the goal via assuming its negation to derive impossibility.

Do NOT count ordinary forward algebra unless it is clearly framed as working from the goal backward. Return only the number between `<count>` and `</count>`. If none, return `<count>0</count>`.

---

## C.2 MAIN EXPERIMENT

In this section, we provide training details not mentioned in the main text.

**Base Model** Our main experiments are conducted using Qwen2.5-Math-7B, a model designed for mathematical reasoning tasks. Due to the limitation of 8k tokens in the DeepSeek-R1 expert trajectories and the 4096 context length of Qwen2.5-Math-7B, we follow LUFFY's approach and increase the model's rope theta from 10000 to 40000.

**Dataset** Our data source is OpenR1-Math-46k-8192 (Yan et al., 2025), which filters out incorrect trajectories and those with token lengths greater than 8192 from OpenR1-Math-200k (Face, 2025). Since OpenR1-Math-200k contains at least two trajectories generated by Deepseek-R1 for each problem, we also collect the corresponding second trajectory for each entry in OpenR1-Math-46k-8192, resulting in approximately 46k prompts and 92k trajectories. Furthermore, since we did not require the model to split the output into think and answer parts in the system prompt, we retain only the tokens between the `<think>` and `</think>` tags from the original Deepseek-R1 output as candidate trajectories. During micro-group sampling, we first calculate the corresponding token position based on the desired ratio, then locate the first reasoning step delimiter (typically a double newline in the Deepseek-R1 style) after this position to ensure the atomicity and completeness of the reasoning steps.

**SFT** For all SFT methods, we train on the 92k <prompt, trajectory> pairs for 2 full epochs as described above. The hyperparameters are set as follows: sequence length of 16,384, learning rate of 5e-5 with a warmup ratio of 0.1, and a batch size of 32. The remaining experimental settings are consistent with those used in OpenR1-Qwen-7B (Face, 2025). All SFT training is conducted using the OpenRLHF (Hu et al., 2024) framework.

**RL** All RL methods use a variant of GRPO, Dr.GRPO (Liu et al., 2025c), which removes length normalization and the standard deviation regularization in advantage calculation. The hyperparameters are as follows: batch size of 128, 8 rollouts per group, max prompt length of 1024, max response length of 8192, and a constant learning rate of 1e-6. We use vLLM (Kwon et al., 2023) for rollouts during training with a sampling temperature set to 1.0. RL training is conducted using the verl (Sheng et al., 2025) framework.

**TRAPO** Since the dataset contains only two expert trajectories per prompt, and we provide expert guidance for up to 4 out of 8 rollouts in each group, we randomly select one expert trajectory as a potential guidance source for each micro-group to ensure diversity in the expert data. Except for setting the trust region parameter $\alpha$ to 0.1, all other training parameters are consistent with those used in RL and are implemented within the verl framework.

**Evaluation** For evaluation, we use a lower temperature of 0.6 and a maximum generation length of 8,192 tokens. We employ Math-verify and OAT-Grader (Liu et al., 2025b) frameworks for evaluation.

**Prompt Template** Following SimpleRL (Zeng et al., 2025), we use the simplest prompt template across all methods and models to ensure that the improvement in reasoning capability comes solely from the training. The template is as follows:

> **Prompt Template**
>
> <|im_start|>system
> You are a helpful assistant.
> <|im_end|>
>
> <|im_start|>user
> {problem}
> Please reason step by step, and put your final answer within \boxed{}.
> <|im_end|>

## D  HYPERPARAMETER STUDY

### D.1  ABLATION OF TRUST-REGION PARAMETER

We apply equations 2 and 3 to a single token, which leads to the following form:

$$\nabla_\theta L = \frac{1}{p_T^\theta(y_n|\boldsymbol{x}, \boldsymbol{y}_{<n}) \ \text{or} \ \max\left(p_T^\theta(y_n|\boldsymbol{x}, \boldsymbol{y}_{<n}), \alpha\right)} \nabla_\theta p_T^\theta(y_n|\boldsymbol{x}, \boldsymbol{y}_{<n}). \tag{14}$$

The weight (the fractional factor) in front of the gradient term varies with $p_T^\theta(y_n|\boldsymbol{x}, \boldsymbol{y}_{<n})$ as shown in Figure 7(a). When $\alpha = 0$, TrSFT is equivalent to SFT, meaning that all tokens lie within the trust region. As $\alpha$ increases, the gradient update weight of tokens outside the trust region is reduced to a fixed value of $1/\alpha$, significantly mitigating the disruptive impact of large gradient updates from these tokens on the model parameters. However, when $\alpha$ becomes too large, all tokens are assigned the same gradient weight, completely losing the characteristic of SFT where tokens with lower probability require more learning. Therefore, choosing an appropriate value of $\alpha$ is crucial to balance the resistance against low-probability tokens and the need to effectively learn from them. Figure 7(b) shows the training performance of TRAPO under different values of $\alpha$ (0, 0.05, 0.1, 0.2, 0.5, 1). We observe that the best performance is achieved when $\alpha = 0.1$, with both higher and lower values of $\alpha$ failing to reach the optimal performance.

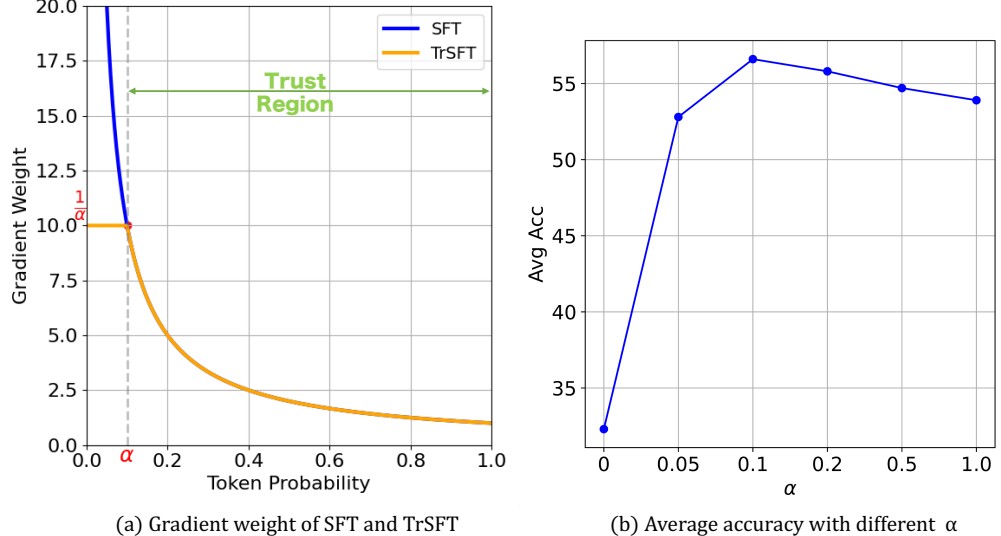

(a) Gradient weight of SFT and TrSFT    (b) Average accuracy with different $\alpha$

Figure 7: (a) The plot showing the change in gradient weight with respect to token probability in SFT and TrSFT. (b) The average accuracy on five mathematical reasoning benchmarks for Qwen2.5-Math-7B trained with TRAPO for different values of $\alpha$.

## D.2 ABLATION OF MICRO-GROUP HYPERPARAMETERS

In micro-group sampling, we adopt an adaptive strategy for determining expert prefix length. Unlike fixed-prefix approaches, this dynamic mechanism selects an appropriate amount of expert guidance per prompt, providing stronger assistance when unguided rollouts perform poorly.

Each micro-group $g_i$ contains several tunable hyperparameters: (1) the prefix-length proportion $L_i$, (2) the reward threshold $t_i$, and (3) the number and sizes of micro-groups. All variants were trained on Qwen2.5-Math-7B for 100 steps, with average accuracy reported across six mathematical reasoning benchmarks.

**1. Prefix-length proportion.**

Table 3: Ablation of the prefix-length proportion.

| Variant | $L_1$ | $L_2$ | $L_3$ | $L_4$ | Avg. Perf. |
|---------|-------|-------|-------|-------|------------|
| A | 0 | 0.2 | 0.5 | 1.0 | 39.4 |
| B | 0 | 0.2 | 0.5 | 0.7 | 37.6 |
| C | 0.1 | 0.2 | 0.5 | 1.0 | 34.1 |
| D | 0 | 0.1 | 0.8 | 1.0 | 39.2 |
| E | 0 | 0.3 | 0.4 | 1.0 | 39.5 |

As shown in Table 3, compared with Variant A, Variants B and C confirm the importance of our two design principles: (1) the first micro-group must receive no expert prefix to preserve unguided exploration, and (2) the final micro-group should expose the full expert trajectory. When modifying only the intermediate prefix proportions (Variants D and E), performance remains largely unchanged, suggesting that TRAPO is robust to these internal ratios.

**2. Reward threshold.**

We maintain a monotonically increasing threshold schedule to match the naturally increasing expected reward of longer prefixes. As shown in Table 4, constant thresholds (B) or decreasing schedules (C) reduce performance. As long as thresholds increase in a reasonable range (D and E), their exact values have minimal effect.

Table 4: Ablation of the reward threshold.

| Variant | $t_1$ | $t_2$ | $t_3$ | $t_4$ | Avg. Perf. |
|---------|-------|-------|-------|-------|------------|
| A | -1 | 0.5 | 0.7 | 0.9 | 39.4 |
| B | -1 | 0.5 | 0.5 | 0.5 | 38.1 |
| C | -1 | 0.9 | 0.7 | 0.5 | 37.4 |
| D | -1 | 0.55 | 0.75 | 0.95 | 38.9 |
| E | -1 | 0.45 | 0.65 | 0.85 | 39.3 |

**3. Number and sizes of micro-groups.**

Table 5: Ablation of the number and sizes of micro-groups.

| Variant | $(n_1, n_2, \ldots)$ | Avg. Perf. |
|---------|----------------------|------------|
| A | (4, 2, 1, 1) | 39.4 |
| B | (2, 4, 1, 1) | 37.2 |
| C | (4, 4) | 38.0 |
| D | (4, 3, 1) | 39.1 |

For the organization of micro-groups, two structural principles are crucial: (1) the first unguided micro-group must be sufficiently large to support exploration, and (2) at least one intermediate group should provide partial guidance to bridge unguided and fully guided rollouts. As shown in Table 5, shrinking the unguided group (B) or removing intermediate groups (C) harms performance. Configurations satisfying both principles (A and D) achieve similar accuracy.

## E  ADDITIONAL COST AND EFFICIENCY ANALYSIS

This section provides further details on the computational efficiency of TRAPO, supplementing the discussion in the main paper. Due to the longer generation lengths and the sequential execution required by micro-group sampling, TRAPO incurs a higher per-step computational cost compared to standard GRPO. However, under matched wall-clock training-time budgets, TRAPO consistently achieves stronger reasoning performance, even though it executes slightly fewer training steps than the baselines. This suggests that TRAPO benefits from more effective training-time scaling.

**Overall GPU-hour usage.**  Table 6 reports the total GPU hours consumed by each method. For the SFT-then-RL pipeline, this measurement includes both the SFT stage and the subsequent RL stage. We also report the average number of expert-prefix tokens (*Expert Tokens*) and target-model-generated tokens (*Target Tokens*) per RL trajectory.

**Training dynamics under matched wall-clock budgets.**  To assess training-time efficiency, we measure the number of steps completed by each method under equal GPU-hour budgets, together with the mean reward of the collected rollout batches. As reward is strongly correlated with downstream reasoning performance, it serves as a useful indicator of learning progress. For SFT-then-RL, the GPU-hour measurement below begins at the start of the RL stage.

**Summary.**  Across all matched training-time conditions, TRAPO consistently achieves higher reward, which is an indicator of stronger reasoning ability than GRPO and LUFFY, even though it performs fewer training steps. This highlights TRAPO's advantageous sample efficiency and the effectiveness of combining SFT and RL through a trust-region formulation. Moreover, compared to the SFT-then-RL pipeline, TRAPO attains better performance while requiring less total GPU hours overall.

Table 6: GPU hours and token statistics across training methods.

| Model | GPU hours | Expert Tokens | Target Tokens |
|---|---|---|---|
| GRPO | $68 \times 8$ | 0 | 1189 |
| SFT-then-RL | $(42 + 105) \times 8$ | 0 | 4760 |
| LUFFY | $81 \times 8$ | 519 | 2439 |
| TRAPO | $89 \times 8$ | 501 | 2545 |

Table 7: Reward progression under matched GPU-hour budgets.

| GPU hours | GRPO | SFT-then-RL | LUFFY | TRAPO |
|---|---|---|---|---|
| $10 \times 8$ | 0.521 | 0.650 | 0.472 | 0.601 |
| $20 \times 8$ | 0.579 | 0.712 | 0.594 | 0.684 |
| $30 \times 8$ | 0.626 | 0.726 | 0.665 | 0.744 |
| $40 \times 8$ | 0.621 | 0.710 | 0.697 | 0.722 |
| $50 \times 8$ | 0.652 | 0.732 | 0.675 | 0.768 |

## F   PREFIX USAGE DYNAMICS DURING TRAINING

To better understand how the adaptive prefix-selection mechanism behaves during training, we track the number of trajectories receiving expert prefix guidance at each training step. The results are shown in Figure 8. Two key phenomena emerge:

- **Unguided trajectories consistently dominate across the entire training process.** This ensures that the target model maintains strong self-exploration capability and remains aligned with the evaluation setting, which uses no expert prefixes at test time.

- **The number of expert-guided trajectories decreases steadily as training progresses.** Early in training, the model frequently falls below the reward thresholds and therefore benefits from partial or full expert-prefix guidance. As performance improves, fewer trajectories trigger guidance, reflecting that the model becomes increasingly capable of solving problems without external intervention.

This pattern confirms that TRAPO's adaptive mechanism realizes the intended behavior: it provides expert prefixes only when helpful, and its reliance on expert data naturally diminishes as the target model improves. This demonstrates that TRAPO leverages *smarter* guidance rather than *more* guidance, and avoids over-conditioning on expert trajectories.

## G   CASE STUDY

To intuitively demonstrate the differences in style and problem-solving ability of models trained with different methods, we select a polynomial factorization problem from the MATH-500 benchmark as a case study. As shown in Figure 9, the model trained with GRPO exhibits a consistent forward-reasoning style in problem solving, also retaining the base model's tendency to employ code verification at the final step. However, due to limited self-reflection and self-correction abilities, it is prone to simple computational and logical mistakes. In contrast, the model trained with SFT demonstrates a rigid imitation of DeepSeek-R1, often engaging in excessive reasoning that exceeds the maximum sequence length budget. The model trained with TRAPO not only incorporates the verification and reflective reasoning patterns observed in expert trajectories, but also preserves the base model's ability to provide concise and clear planning before each reasoning step (e.g., outlining "Step 1 / Step 2 / Step 3 . . . "). This demonstrates that TRAPO effectively balances the use of expert data with the exploration of its own solution space, combining the strengths of both to maximize training efficiency and reasoning performance.

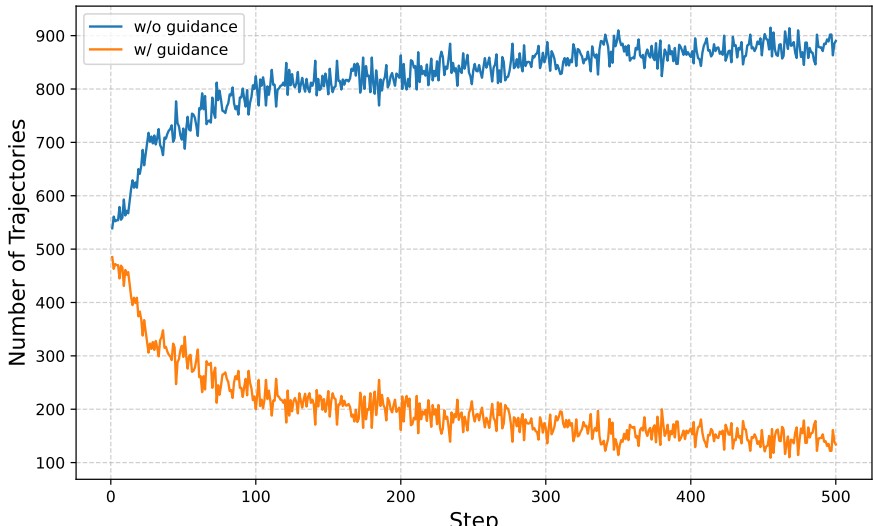

Figure 8: Dynamics of prefix usage across training steps. We plot the number of trajectories that receive expert-prefix guidance versus those that remain unguided.

## H  THE USE OF LARGE LANGUAGE MODELS (LLMS)

We ensure that our use of LLMs in this work fully complies with the ICLR 2026 policy. All research ideas and scientific contributions originate solely from the human authors, with no involvement of LLMs in research ideation. In addition, LLMs were not used to retrieve or discover related works. Their usage was strictly limited to language polishing and to the automated counting of reasoning behaviors reported in Appendix C.1. All generated content was carefully reviewed by the authors to ensure it faithfully reflects our original intent.

**Question**

The polynomial $x^3 - 3x^2 + 4x - 1$ is a factor of $x^9 + px^6 + qx^3 + r$. Enter the ordered triple $(p, q, r)$.
Answer: (6,31,-1)

**GRPO**

[Problem decomposition]
To solve the problem, we need to determine the coefficients p, q, and r such that the polynomial $x^9 + px^6 + qx^3 + r$ is divisible by $x^3 - 3x^2 + 4x - 1$. This means that if $f(x) = x^3 - 3x^2 + 4x - 1$, then $f(x)$ divides $x^9 + px^6 + qx^3 + r$. In other words, the roots of $f(x)$ must also be roots of $x^9 + px^6 + qx^3 + r$ …
[Method choice / plan]
Let's denote the roots of $x^3 - 3x^2 + 4x - 1 = 0$ by \(a\), \(b\), and \(c\)...
[Forward algebra (power reduction)]
We need to express \(a^9\), \(b^9\), and \(c^9\) in terms of \(a^3\), \(b^3\), and \(c^3\), and then in terms of \(a\), \(b\), and \(c\).
[Constraint extraction]
First, let's find \(a^6\): … Now, let's find \(a^9\):…Combining like terms, we get:…We can solve these equations step by step. First, solve the first two equations for \(p\) and \(q\):
[Computation error]
So the ordered triple \((p, q, r)\) is $(\frac{493}{73}, \frac{19829}{219}, \frac{1418}{219})$.
[Verification]
Let's verify this solution using Python to ensure accuracy.
```python…

**SFT**

[Recall knowledge]
Hmm, okay. Let me start by recalling that if a polynomial \(A(x)\) is a factor of another polynomial \(B(x)\), then all the roots of \(A(x)\) are also roots of \(B(x)\)…
[Overthink]
But since \(a\), \(b\), \(c\) are roots of \(x^3 - 3x^2 + 4x - 1 = 0\)…
Alternatively, maybe we can use polynomial division…
But polynomial division might be tedious here. Maybe there's a smarter way. Let me think…
Wait, let's re-examine. For the \(x^7\) term:…
Wait, the \(x^6\) terms come from:…
But we don't know \(c\) yet. Let's continue…
Wait, no. Wait, when multiplying \(S(x)\) by \((x^3 - 3x^2 + 4x - 1)\), the constant term…
Wait, let me do each term:…
Wait, perhaps we can use equation 7 or equation 6 again? Wait, but we already used them. Wait, perhaps we need another equation. Wait, but all equations have been used. Hmm. Wait, maybe we need to use the original polynomial equation…
Wait, no. Wait, actually, if \(x^9 + p x^6 + q x^3 + r =0\), then substituting the expression for \(x^9\) from above:…

**TRAPO**

[Clear Plan and Sufficient and moderate contemplation]
Sure, I can help you solve this problem. The goal is to find the ordered triple \((p, q, r)\) such that the polynomial \(x^3 - 3x^2 + 4x - 1\) is a factor of \(x^9 + px^6 + qx^3 + r\).
### Step 1: Understand the Problem
Given that \(x^3 - 3x^2 + 4x - 1\) is a factor of \(x^9 + px^6 + qx^3 + r\), this implies that when we divide …
### Step 2: Find the Roots of the Cubic Polynomial
First, let's find the roots of the cubic polynomial \(x^3 - 3x^2 + 4x - 1 = 0\). This might be a bit tricky,…
### Step 3: Polynomial Division or Substitution
Since \(x^3 - 3x^2 + 4x - 1\) is a factor of \(x^9 + px^6 + qx^3 + r\), we can use polynomial division or substitute-ion. However, polynomial division might be cumbersome for degree 9. Instead, let's use substitution…
### Step 4: Substitute and Simplify
Let's compute \(y^3 + py^2 + qy + r\) with \(y = 3x^2 - 4x + 1\)…
### Step 5: Express Higher Powers in Terms of Lower Powers
We need to express \(x^9\) in terms of \(x^2\), \(x\), and constants using the relation \(x^3 = 3x^2 - 4x + 1\)…
### Step 6: Substitute into the Original Polynomial
Now, substitute these expressions into the original polynomial \(x^9 + px^6 + qx^3 + r\):…
### Step 7: Solve the System of Equations
First, let's solve the first two equations for \(p\) and \(q\)…
### Step 8: Verify the Solution
To ensure that our solution is correct, we can verify by substituting \(p = 6\), \(q = 31\), \(r = -1\) back into the original polynomial and checking if \(x^3 - 3x^2 + 4x - 1\) divides it.
### Conclusion
The ordered triple \((p, q, r)\) is \((6, 31, -1)\). Therefore, the answer is \(\boxed{(6, 31, -1)}\).
### Reflection
This problem required understanding polynomial division, substitution, and solving a system of equations. The key was to express higher powers of \(x\) in terms of lower powers using the given cubic …

Figure 9: A case study on the MATH-500 benchmark: polynomial factorization problem.

