# OpenReview forum: "Trust-Region Adaptive Policy Optimization"
_ICLR.cc/2026/Conference — ICLR 2026 Poster_

### Official Review · Reviewer_3vHn · 2025-10-30

**Soundness:** 3
**Presentation:** 3
**Contribution:** 3
**Rating:** 8
**Confidence:** 3

**Summary:**

This paper introduces TRAPO (Trust-Region Adaptive Policy Optimization), a novel one-stage hybrid framework that unifies Supervised Fine-Tuning (SFT) and Reinforcement Learning (RL) for post-training large language models (LLMs). The authors argue that the conventional two-stage pipeline—first SFT, then RL—creates a mismatch: SFT enforces rigid imitation that suppresses exploration and induces catastrophic forgetting, thereby weakening RL’s learning potential.

To overcome this, TRAPO interleaves SFT and RL within each training instance. It performs:
- SFT loss on expert prefixes, to absorb expert reasoning patterns.
- RL loss on model rollouts, to maintain exploration and self-improvement.

A key innovation is Trust-Region SFT (TrSFT), which minimizes forward KL divergence inside a trust region but attenuates optimization outside, effectively shifting toward reverse KL to encourage mode-seeking rather than mode-covering. This ensures stable and meaningful updates beneficial for RL. An adaptive prefix-selection mechanism dynamically adjusts expert prefix length according to model performance, balancing guidance with exploration.

Experiments across five mathematical reasoning benchmarks (AIME2024, AMC, MATH-500, Minerva, OlympiadBench) show TRAPO outperforms SFT, RL, and the SFT-then-RL pipeline by +2–6 points, and achieves the highest average accuracy of 56.6%. TRAPO also generalizes well to non-mathematical tasks (ARC-c, MMLU-Pro) and exhibits superior test-time scaling properties.

**Strengths:**

1. Novel Conceptual Integration
The paper’s central insight—recasting SFT and RL as interleaved processes at the instance level—is original and well-motivated. By linking expert prefix imitation with RL rollouts in a trust region, TRAPO presents a principled path toward more synergistic post-training.

2. Theoretical Clarity
The authors diagnose a specific instability—distribution blending due to the mode-covering nature of forward KL—and formally derive how the clipped gradient in TrSFT converts the optimization dynamics toward reverse KL’s mode-seeking behavior. This analysis provides strong theoretical grounding rarely seen in empirical LLM post-training work.

4. Comprehensive Evaluation

The paper offers convincing empirical validation:
- Clear improvement over both pure RL baselines (GRPO, PRIME-Zero, etc.) and hybrid methods (LUFFY, ReLIFT).
- Extensive ablation studies show the distinct contribution of micro-group sampling and TrSFT.

**Weaknesses:**

1. Limited Scope of Evaluation

All major experiments focus on mathematical reasoning benchmarks. While the generalization test on ARC-c and MMLU-Pro is included, more diverse domains (e.g., commonsense reasoning, coding, dialogue) would better validate TRAPO’s universality.

2. Lack of Cost and Efficiency Analysis

TRAPO’s training procedure—especially with multiple micro-groups and prefix sampling—appears more computationally expensive than standard RL. The paper does not quantify overhead (e.g., training time, token cost, GPU hours), which is important for practical adoption.

3. Trust-Region Parameter Sensitivity

Although the trust-region boundary parameter α is analyzed in the appendix, the main paper provides limited discussion on its tuning sensitivity and stability under different model scales.

**Questions:**

None

---

> ### Author Response · Authors · 2025-11-23
> **Response to Reviewer 3vHn**
>
> We are deeply grateful for the reviewer’s generous and thoughtful assessment! The recognition of TRAPO’s conceptual motivation, its theoretical grounding, and the breadth of the empirical analysis is highly encouraging. We especially appreciate the reviewer’s careful reading of the trust-region formulation, the SFT–RL integration, and the role of adaptive prefixes, which are central aspects of the work, and we are glad that they came through clearly. The reviewer also identifies important directions for strengthening the paper, particularly regarding broader domain coverage, training efficiency, and the sensitivity of the trust-region parameter. We address these points in detail below.
>
> **[Q1]**: Limited Scope of Evaluation.
>
> **[A1]**: We thank the reviewer for the helpful suggestion regarding broader evaluation scope! Following the reviewer’s recommendation, we further extended our study to a code-generation setting using the HumanEval benchmark, which consists of 164 Python coding problems. Under identical evaluation settings with greedy decoding and a maximum generation length of 8192 tokens, we compared TRAPO with GRPO, SFT-then-RL, and LUFFY. TRAPO achieves the highest pass@1 performance among all methods, indicating that its benefits are not limited to mathematical reasoning but also transfer to domains that differ substantially from the mathematical setting.
>
> | Dataset   | Base  | GRPO  | SFT   | SFT-then-RL | LUFFY | TRAPO |
> | --------- | ----- | ----- | ----- | ----------- | ----- | ----- |
> | HumanEval | 79.27 | 94.51 | 16.46 | 32.48       | 69.51 | 95.12 |
>
> **[Q2]**: Lack of Cost and Efficiency Analysis.
>
> **[A2]**: We thank the reviewer for raising this important point! We have added a detailed cost-and-efficiency analysis in **[CA1]**, where we compare TRAPO with all baselines under matched wall-clock training budgets. Although TRAPO introduces additional computation per step due to sequential micro-group rollouts and longer generation length, the overall overhead is modest. In practice, the per-step slowdown is dominated by generation length rather than by the micro-group mechanism itself.
>
> More importantly, when training time is controlled to be the same across methods, TRAPO consistently achieves higher reasoning performance than GRPO, LUFFY, and SFT-then-RL. Compared with the two-stage SFT-then-RL pipeline, TRAPO not only reaches better final accuracy but also requires fewer total GPU hours to do so. These results demonstrate that the computational overhead of micro-groups is small relative to the performance gains they provide, and that TRAPO is competitive in both efficiency and effectiveness for practical adoption.
>
> **[Q3]**: Trust-Region Parameter Sensitivity.
>
> **[A3]**: We acknowledge the reviewer’s point that the trust-region parameter $\alpha$ plays a central role in TRAPO. As described in Appendix D, we use $\alpha$ = 0.1 as the default setting for all experiments. The value of $\alpha$ directly governs how strongly expert prefixes influence the target model during training, and adjusting it is an important part of deploying TRAPO in new settings.
>
> In practice, $\alpha$ can be tuned using the entropy dynamics of generated tokens as an effective diagnostic. When the entropy drops too quickly, which indicates that RL is collapsing into overly deterministic policies, decreasing $\alpha$ allows TrSFT to internalize expert information more aggressively, which counteracts entropy collapse and stabilizes learning. Conversely, if the entropy plateaus or even increases unexpectedly, increasing $\alpha$ narrows the trust region and prevents the model from relying too heavily on expert trajectories. This provides a simple and reliable tuning strategy by monitoring entropy over training and decreasing $\alpha$ when RL becomes overly deterministic, while increasing $\alpha$ when expert influence is too strong.

---

> ### Author Response · Authors · 2025-11-24
> **Response to Reviewer 3vHn ([A1] Follow-Up)**
>
> We appreciate the reviewer’s follow-up remarks and would like to address the two questions below.
>
> 1. For SFT, SFT-then-RL and LUFFY, we observe degradation primarily because the Qwen2.5-Math-7B base model already possesses reasonable code-generation ability, and even uses code to validate mathematical reasoning. Training these methods purely on long-form mathematical CoT trajectories overwrites this native coding behavior. In HumanEval, most failures arise not from logical mistakes but from disrupted output structure such as repetitive thinking leading to truncation, or outputs no longer forming valid code blocks that the evaluator can parse. In short, these methods degrade because they overwrite the model’s original code-generation mode.
>
> 2. In contrast, GRPO improves because it relies solely on self-generated rollouts, refining patterns the base model already knows without introducing conflicting external trajectories. This avoids catastrophic forgetting and further concentrates probability mass on correct coding patterns. TRAPO goes one step further as TrSFT internalizes expert reasoning while preserving the base model’s behavioral structure. By clipping gradient weights for low-probability expert tokens, TrSFT limits the impact of expert tokens that diverge from the target model’s preferred patterns, preventing harmful updates to its coding capability. As illustrated by the case study in Appendix G of the updated manuscript, TRAPO preserves some characteristic output styles of the base model while enabling deeper reasoning before generating the final answer. These observations confirm that TRAPO provides the right balance between integrating expert knowledge and maintaining the model’s inherent strengths.

---

### Official Review · Reviewer_Ux2e · 2025-10-31

**Soundness:** 3
**Presentation:** 3
**Contribution:** 3
**Rating:** 6
**Confidence:** 4

**Summary:**

This paper proposes TRAPO, a one-stage post-training framework that interleaves SFT and RL at the instance level. For each prompt, the model receives an expert prefix (possibly empty), then rolls out its own completion; training applies TrSFT (a trust-region variant of SFT) on the expert tokens and an RL loss on the self-generated suffix. The authors argue TrSFT curbs the mode-covering tendency of forward-KL SFT by clipping per-token gradient weights outside a probability "trust region," effectively shifting behavior toward mode-seeking (reverse-KL-like) updates that are friendlier to RL. They also introduce micro-group sampling, which adaptively increases the prefix length only when earlier unguided (or shorter-guided) rollouts underperform. Empirically, on five math-reasoning benchmarks, TRAPO outperforms SFT, pure RL, and the common SFT-then-RL pipeline. On average it improves over SFT and GRPO by and beats SFT-then-RL.

**Strengths:**

1. The per-instance coupling, SFT on prefixes + RL on suffixes, neatly addresses the two-stage inconsistency the authors highlight. This method directly targets the lack of exploration space for SFT'ed policies, which is a concrete and important problem in RL today. The micro-group schedule is clear and practical.
2. Clipping the token-level gradient weight with a threshold α seems to be an intuitive way to prevent outsized updates on low-probability expert tokens. Relevant hyperparameter study is inlcuded.
3. The pseudocode for the adaptive sampling and the combined gradient update is helpful for me to understand the method in general.

**Weaknesses:**

1. One (or a type of) baseline I consider missing from the paper is existing work that directly improve sampling diversity during rollout in RL. Simple baseline might include increasing temperature, or other algorithms that encourage diverse samples. I recognize the the authors' argument that SFT may decrease sampling diversity, but there is also simple entropy control mechanisms like [1] (and possibly some others).
2. Training relies on OpenR1-Math expert trajectories (DeepSeek-R1), and the core evaluations are all math-reasoning; stronger non-math results would better support claims of broader reasoning gains from TRAPO.
3. The RL baseline is implemented with a GRPO variant and no KL penalty; the paper should clarify whether this favors TRAPO (which already regularizes via TrSFT).

[1] He, Jujie, et al. "Skywork open reasoner 1 technical report." arXiv preprint arXiv:2505.22312 (2025).

**Questions:**

1. If you add back a standard KL penalty to the RL objective, how do performance and stability change relative to TrSFT alone?
2. How does TRAPO perform if the expert prefixes come from a weaker trajectory model, or from noisy/partial solutions? Does TrSFT still help, or does it overfit to poor guidance?
3. Does the dynamic prefix selection strategy incur computation overhead? If so, by how much?
4. Can you show ablations on code or multimodal reasoning, where the expert prefixes may differ stylistically from the target model's distribution?
5. You randomly pick one expert trajectory per micro-group. Would prioritizing trajectories (e.g., by diversity or difficulty) further help?

---

> ### Author Response · Authors · 2025-11-23
> **Response to Reviewer Ux2e [Part 1/3]**
>
> We appreciate the reviewer’s thoughtful and constructive assessment! The comments provide a clear and balanced perspective on both the methodological contributions and the remaining gaps in evaluation and comparison. We are encouraged that the reviewer found the instance-level SFT–RL coupling, the rationale behind TrSFT, and the structure of the micro-group mechanism clearly presented. The points raised regarding diversity-oriented baselines, broader non-math evaluation and the clarification regarding the connection between TrSFT and the KL penalty are very helpful, and we address them in detail below.
>
> **[Q1]**: One (or a type of) baseline I consider missing from the paper is existing work that directly improves sampling diversity during rollout in RL. Simple baseline might include increasing temperature, or other algorithms that encourage diverse samples. I recognize the  authors' argument that SFT may decrease sampling diversity, but there are also simple entropy control mechanisms like [1] (and possibly some others).
>
> **[A1]**: We thank the reviewer for suggesting additional baselines that explicitly increase sampling diversity during RL rollouts! The standard two-stage SFT-then-RL pipeline suffers from two inherent limitations: (1) SFT tends to overfit the training distribution, which restricts exploration during RL, and (2) SFT forces the target model to rigidly imitate the expert trajectories, often causing catastrophic forgetting of the model’s useful prior knowledge. Methods that solely increase sampling diversity during RL (e.g., higher temperature or stronger entropy regularization) primarily address issue (1) but do not mitigate issue (2). This is why we did not initially include such baselines. For the reviewer’s reference, our main experiments indeed employed a sampling temperature of $\tau = 1.0$ and an entropy loss coefficient of $\lambda_e = 0.001$ across all RL methods to encourage exploration. Following the reviewer’s suggestion, we additionally evaluated baselines with $\tau$ increased to 1.2 and $\lambda_e$ increased to 0.005, and compared them directly against TRAPO. The results are as follows:
>
> | Model                                            | AIME2024 | AIME2025 | AMC  | MATH-500 | Minerva | Olympiad | Avg. |
> | ------------------------------------------------ | -------- | -------- | ---- | -------- | ------- | -------- | ---- |
> | TRAPO ($\tau = 1.0$, $\lambda_e = 0.001$)        | 28.3     | 23.9     | 66.2 | 89.2     | 41.5    | 57.6     | 51.2 |
> | SFT-then-RL  ($\tau = 1.0$, $\lambda_e = 0.001$) | 33.5     | 23.1     | 62.3 | 86.6     | 41.2    | 47.6     | 49.1 |
> | SFT-then-RL  ($\tau = 1.2$, $\lambda_e = 0.001$) | 30.9     | 23.1     | 65.6 | 87.0     | 40.8    | 48.8     | 49.3 |
> | SFT-then-RL  ($\tau = 1.2$, $\lambda_e = 0.005$) | 29.6     | 22.3     | 66.8 | 89.0     | 38.6    | 50.2     | 49.4 |
>
> The results show that TRAPO outperforms both diversity-enhanced baselines, indicating that merely increasing sampling diversity during RL does not bridge the gap, and that addressing the SFT–RL inconsistency remains essential for achieving stronger reasoning performance.

---

> > ### Author Response · Authors · 2025-11-23
> > **Response to Reviewer Ux2e [Part 2/3]**
> >
> > **[Q2]**: If you add back a standard KL penalty to the RL objective, how do performance and stability change relative to TrSFT alone?
> >
> > **[A2]**: We removed the KL penalty term in all RL baselines to minimize confounding method-specific factors and to remain consistent with the LUFFY baseline. In standard GRPO, the KL penalty keeps the policy close to a reference model (typically the initial target model), and a large coefficient $\beta$ can overly constrain exploration and harm reasoning performance. In contrast, TrSFT plays a different role: it is designed to internalize the expert model’s reasoning behavior while preserving the useful structure of the target model’s existing reasoning patterns. TrSFT is therefore not intended to restrict the target model around a fixed reference point, and its regularization effect is qualitatively distinct from that of the KL penalty. Following the reviewer’s suggestion, we additionally incorporated a KL penalty term with coefficient $\beta=0.01$ on top of TRAPO, and trained this variant for 100 RL steps. The evaluation results are reported below:
> >
> > | Model                | AIME2024 | AIME2025 | AMC  | MATH-500 | Minerva | Olympiad | Avg. |
> > | -------------------- | -------- | -------- | ---- | -------- | ------- | -------- | ---- |
> > | TRAPO w/ KL penalty  | 16.7     | 11.4     | 44.2 | 72.6     | 30.2    | 43.6     | 36.5 |
> > | TRAPO w/o KL penalty | 17.1     | 12.7     | 48.5 | 76.8     | 33.5    | 47.7     | 39.4 |
> >
> > The results suggest that adding a KL penalty offers no benefit and may interfere with the intended effect of TrSFT, possibly because the two regularizers pull the policy in incompatible directions.
> >
> > **[Q3]**: How does TRAPO perform if the expert prefixes come from a weaker trajectory model, or from noisy/partial solutions? Does TrSFT still help, or does it overfit to poor guidance?
> >
> > **[A3]**: TRAPO remains robust even when expert prefixes come from weaker trajectory models or contain noisy or partial solutions. This robustness comes from the design of TrSFT, whose objective is to internalize expert knowledge in a stable manner during RL while preserving the useful reasoning structure already present in the target model. Rather than aggressively imitating every expert token, TrSFT applies trust-region clipping on the per-token gradient weights. With the parameter $\alpha$, low-probability expert tokens, typically produced by weak or erroneous prefixes, receive heavily down-weighted updates. As a result, the model does not overfit to poor guidance, and the RL optimization is protected from collapsing into suboptimal behaviors.
> >
> > In contrast, the standard SFT-then-RL pipeline first forces the model to fully mimic all expert trajectories, including incorrect or low-quality ones, which can overwrite prior knowledge and lead to degraded convergence during RL. TrSFT avoids this failure mode by ensuring that only high-probability and thus more reliable expert signals exert strong influence, while noisy or partial prefixes contribute only minimally. Therefore, TrSFT continues to help even under weaker or noisy supervision, rather than overfitting to poor guidance.
> >
> > To verify this, we used Qwen2.5-1.5B-Instruct to generate one trajectory per training problem, yielding 45,792 trajectories in total. Their quality varies widely, with Math-Verify marking 12401 among them as correct and 33391 as incorrect. We then trained both TRAPO and the standard SFT-then-RL baseline on this noisy dataset. The results show that TRAPO remains more robust under noisy supervision than the SFT-then-RL baseline:
> >
> > | Model               | AIME2024 | AIME2025 | AMC  | MATH-500 | Minerva | Olympiad | Avg. |
> > | ------------------- | -------- | -------- | ---- | -------- | ------- | -------- | ---- |
> > | TRAPO (noisy)       | 10.7     | 7.3      | 38.6 | 70.6     | 26.0    | 37.7     | 31.8 |
> > | SFT-then-RL (noisy) | 8.2      | 5.4      | 36.2 | 70.2     | 23.4    | 34.3     | 29.6 |

---

> > > ### Author Response · Authors · 2025-11-23
> > > **Response to Reviewer Ux2e [Part 3/3]**
> > >
> > > **[Q4]**: Does the dynamic prefix selection strategy incur computation overhead? If so, by how much?
> > >
> > > **[A4]**: We provide a systematic analysis of TRAPO’s training-time efficiency in **[CA1]**, where we compare TRAPO with all baselines under matched wall-clock budgets. Although dynamic prefix selection introduces a small amount of per-step overhead, mainly due to sequential execution of micro-groups, the overall impact is modest. In practice, TRAPO’s per-step runtime is only slightly slower than pure GRPO, and the difference is dominated by generation length rather than the adaptive prefix mechanism itself.
> > >
> > > More importantly, when models are compared under the same wall-clock training budget, TRAPO consistently reaches higher reasoning performance than GRPO, LUFFY, and other baselines. Compared with the two-stage SFT-then-RL pipeline, TRAPO not only achieves better final accuracy but also requires less total training time to do so. These results indicate that the computation overhead introduced by dynamic prefix selection is small relative to the performance gains it provides.
> > >
> > > **[Q5]**: Can you show ablations on code or multimodal reasoning, where the expert prefixes may differ stylistically from the target model's distribution?
> > >
> > > **[A5]**: In our main experimental setting, the expert model (DeepSeek-R1) and the target model (Qwen2.5-Math-7B) already differ substantially in output style: R1 typically produces long, elaborate chains of thought, whereas Qwen2.5-Math-7B follows a much shorter reasoning pattern. This stylistic mismatch is precisely where TrSFT is designed to help. By clipping per-token gradient weights with the trust-region parameter $\alpha$, TrSFT balances the influence of expert prefixes that differ from the target model’s native distribution against the model’s own RL-driven refinements, preventing over-imitation of stylistically incompatible guidance.
> > >
> > > We therefore expect TRAPO to remain effective in other reasoning domains where the expert data deviate stylistically from the target model. Due to time and resource constraints, we conducted a lightweight evaluation on code generation, where style mismatch is also pronounced. We used the widely adopted HumanEval benchmark (164 problems), with a maximum generation length of 8192 and greedy decoding, and report Pass@1 results below:
> > >
> > > |           | Base  | GRPO  | SFT   | SFT-then-RL | LUFFY | TRAPO |
> > > | --------- | ----- | ----- | ----- | ----------- | ----- | ----- |
> > > | HumanEval | 79.27 | 94.51 | 16.46 | 32.48       | 69.51 | 95.12 |
> > >
> > > **[Q6]**: You randomly pick one expert trajectory per micro-group. Would prioritizing trajectories (e.g., by diversity or difficulty) further help?
> > >
> > > **[A6]**: We thank the reviewer for the insightful suggestion regarding trajectory prioritization! In our current micro-group sampling design, for each prompt we randomly select one expert trajectory to provide the prefix guidance, rather than relying on a fixed trajectory per prompt. This randomness ensures broader exposure to expert behavior despite the limited number of trajectories available per problem.
> > >
> > > Difficulty is inherently tied to the prompt and cannot be controlled by trajectory selection. However, diversity is indeed a meaningful criterion for sampling from the trajectory pool. We expect that prioritizing diverse trajectories, such as ensuring that prefixes for different micro-groups originate from non-overlapping expert trajectories, could provide the target model with a richer view of the expert’s reasoning patterns and further improve the internalization of expert knowledge.
> > >
> > > Unfortunately, the original OpenR1-Math-200k dataset contains only two trajectories for most prompts, and due to time and computational constraints we were unable to explore diversity-aware sampling in this submission. We agree that this is a promising direction, and we plan to investigate it more thoroughly in future work.

---

> > > > ### Comment · Reviewer_Ux2e · 2025-11-25
> > > >
> > > > I appreciate the authors' efforts during the rebuttal and the additional experiments that addressed my concerns. I have carefully reviewed the results above and gone through the revised pdf. I am generally positive about the paper and have decided to maintain my initial rating.

---

### Official Review · Reviewer_Hux5 · 2025-11-01

**Soundness:** 2
**Presentation:** 3
**Contribution:** 2
**Rating:** 4
**Confidence:** 4

**Summary:**

This paper proposes TRAPO, a hybrid framework that unifies external supervision and self-exploration through alternating optimization within each training instance. Concretely, it employs TrSFT, an SFT method that modulates optimization strength via a trust region to eliminate the "mode-covering" defect of standard SFT, and calculates RL loss on the model’s self-generated completions to preserve exploratory ability. Experiments on 5 mathematical reasoning and 2 general-domain reasoning benchmarks demonstrate that TRAPO outperforms standard SFT, pure RL, and SFT-then-RL pipelines.

**Strengths:**

1. The paper is well-organized and easy to follow. The method is clearly defined, and the figures are clear and helpful.

2. The ablation results support the design. Micro-group sampling alone improves over GRPO, while replacing TrSFT with standard SFT causes a clear drop. This suggests the trust region brings stability, and the adaptive prefixes provide useful guidance.

**Weaknesses:**

1. This paper only validates TRAPO’s performance on established benchmarks, including 5 mathematical reasoning benchmarks and 2 general-domain reasoning benchmarks, while excluding newer, harder datasets like AIME25 and GPQA.

2. The micro-group sampling of TRAPO requires sequential processing for each prompt’s micro-groups within a mini-batch. This sequential logic for micro-groups is far less parallelizable than baselines like pure GRPO. Additionally, the training time is not provided, so the tradeoff between TRAPO's performance and efficiency remains unproven.

3. TRAPO relies on multiple hyperparameters that are difficult to tune, including micro-group-specific ones and the TrSFT trust-region parameter alpha. While Figure 7(b) shows accuracy variations of Qwen2.5-Math-7B on mathematical benchmarks with different alpha values, it does not demonstrate stable or optimal performance across parameter adjustment; instead, it reflects that alpha is sensitive to performance, let alone the combined tuning complexity of all hyperparameters.

4. TRAPO feels overengineered: it combines staged micro‑group scheduling with per‑prompt SFT prefix adjustment, which together make the pipeline complex and difficult to control.

**Questions:**

1. How does TRAPO perform on newer, harder benchmarks such as AIME25 and GPQA?

2. How robust is TRAPO to low‑quality or noisy expert trajectories in SFT data? For example, with wrong or meaningless prefixes, does performance degrade gracefully, and what mitigation strategies work best, such as prefix filtering or confidence weighting?

3. How dependent is TRAPO on the Expert Model? What happens if the Expert is weaker or partially misaligned? Specifically, if the Expert Model is identical to the Policy Model, how do performance and stability change?

4. How should these hyperparameters be tuned, such as the number and sizing of micro-groups, the prefix ratios and return thresholds, and the TrSFT trust‑region alpha? Are there recommended defaults or automatic tuning strategies?

---

> ### Author Response · Authors · 2025-11-23
> **Response to Reviewer Hux5 [Part 1/3]**
>
> We sincerely thank the reviewer for the clear and constructive review! We appreciate the reviewer’s positive remarks on the paper’s organization, clarity of method, and supporting ablations. Equally importantly, the reviewer’s comments on benchmark coverage, training efficiency, hyperparameter sensitivity, and system complexity precisely identify the aspects where additional clarification and analysis will strengthen the work. We are grateful for these insights, and we address each point in detail below.
>
> **[Q1]**: How does TRAPO perform on newer, harder benchmarks such as AIME25 and GPQA?
>
> **[A1]** : To more comprehensively assess the model’s capabilities and the effectiveness of our approach, we have further evaluated all relevant methods on two more challenging benchmarks, AIME2025 and GPQA-Diamond, as required by the reviewer. Note that all models evaluated here were trained from Qwen2.5-Math-7B. Since AIME2025 contains fewer samples, we report avg@32, while for GPQA-Diamond we report pass@1. After incorporating the results from both AIME2025 and GPQA-Diamond, we also update the average performance of the models across both six mathematical reasoning benchmarks and three general-domain reasoning benchmarks. The results are as follows:
>
> | Model       | AIME2025(avg@32) | Avg.(Math) | GPQA-diamond(pass@1) | Avg.(Non-Math) |
> | ----------- | ---------------- | ---------- | -------------------- | -------------- |
> | GRPO        | 11.9             | 44.0       | 38.9                 | 55.6           |
> | SFT-then-RL | 23.1             | 49.1       | 23.7                 | 37.6           |
> | ReLIFT      | 20.1             | 47.8       | 37.9                 | 55.5           |
> | LUFFY       | 22.9             | 50.1       | 39.9                 | 57.8           |
> | TRAPO       | 23.9             | 51.2       | 39.4                 | 58.7           |
>
> The latest results confirm that TRAPO achieved satisfactory overall performance on both challenging mathematical and general-domain reasoning benchmarks, surpassing both pure RL methods and RL-with-guidance approaches.
>
> **[Q2]**: The micro-group sampling of TRAPO requires sequential processing for each prompt’s micro-groups within a mini-batch. This sequential logic for micro-groups is far less parallelizable than baselines like pure GRPO. Additionally, the training time is not provided, so the tradeoff between TRAPO's performance and efficiency remains unproven.
>
> **[A2]**: We have provided a detailed analysis of training efficiency for all methods in **[CA1]**, including a comparison of TRAPO and the baselines under matched training-time budgets. While TRAPO has a longer per-step runtime due to the sequential execution of micro-groups and the longer generation lengths, the overall efficiency comparison tells a different story. Under the same wall-clock training budget, TRAPO consistently achieves higher reasoning performance than GRPO, LUFFY, and other baselines. Moreover, compared with the traditional two-stage SFT-then-RL pipeline, which also incorporates expert knowledge, TRAPO not only attains better performance but does so with lower total training cost.

---

> > ### Author Response · Authors · 2025-11-23
> > **Response to Reviewer Hux5 [Part 2/3]**
> >
> > **[Q3]**: How robust is TRAPO to low‑quality or noisy expert trajectories in SFT data? For example, with wrong or meaningless prefixes, does performance degrade gracefully, and what mitigation strategies work best, such as prefix filtering or confidence weighting?
> >
> > **[A3]**: We believe that the design of TrSFT inherently provides robustness to low-quality or noisy expert trajectories. Its objective is to allow the target model to internalize expert knowledge in a stable manner during RL, while preserving the useful portion of the model’s existing capabilities. As a result, when exposed to noisy prefixes, TRAPO does not aggressively imitate low-quality signals; instead, the influence of such trajectories is moderated by the trust-region formulation, which prevents disruptive updates and keeps the RL optimization from collapsing into suboptimal solutions. In contrast, a standard SFT-then-RL pipeline first forces the model to rigidly mimic the noisy expert trajectories, which can overwrite its prior knowledge and lead to degraded RL convergence.
> >
> > To evaluate robustness under noisy supervision, we used Qwen2.5-1.5B-Instruct to generate trajectories for every problem in our original training set, producing a total of 45792 trajectories. These trajectories exhibit substantial quality variance, with Math-Verify labeling 12401 of them as correct and 33391 as incorrect. We trained TRAPO and the standard SFT-then-RL baseline on this noisy dataset, and additionally trained variants using only the correct trajectories to assess the benefit of prefix filtering. The results are as follows:
> >
> > | Model               | AIME2024 | AIME2025 | AMC  | MATH-500 | Minerva | Olympiad | Avg. |
> > | ------------------- | -------- | -------- | ---- | -------- | ------- | -------- | ---- |
> > | TRAPO (noisy)       | 10.7     | 7.3      | 38.6 | 70.6     | 26.0    | 37.7     | 31.8 |
> > | SFT-then-RL (noisy) | 8.2      | 5.4      | 36.2 | 70.2     | 23.4    | 34.3     | 29.6 |
> > | TRAPO (filtered)    | 12.5     | 6.3      | 39.4 | 74.8     | 27.4    | 40.2     | 33.4 |
> >
> > These results demonstrate two key findings: (1) TRAPO is more robust than SFT-then-RL when trained on low-quality trajectories with a mixture of correct and incorrect reasoning, and (2) even a simple prefix-filtering strategy that removes incorrect expert trajectories can yield a meaningful improvement in performance.
> >
> > **[Q4]**: How dependent is TRAPO on the Expert Model? What happens if the Expert is weaker or partially misaligned? Specifically, if the Expert Model is identical to the Policy Model, how do performance and stability change?
> >
> > **[A4]**: We view the expert model primarily as a source of additional structured guidance that TrSFT can internalize in a stable manner during RL. Consequently, stronger expert models naturally lead to larger gains for TRAPO. As shown in **[A3]**, when we use Qwen2.5-1.5B-Instruct as the expert model, whose trajectories have substantially lower average quality than those generated by DeepSeek-R1, the improvements from TRAPO are noticeably smaller, both with and without trajectory filtering. Nevertheless, the results in **[A3]** also demonstrate that TRAPO remains considerably more robust than the standard SFT-then-RL pipeline under low-quality expert supervision. This robustness arises because TrSFT internalizes only the useful parts of the expert signal while preserving the beneficial structure of the target model’s prior knowledge.
> >
> > Regarding the scenario where the expert model is identical to the initial target model, its impact will be identical to  the reference model in standard GRPO. In GRPO, the KL-penalty term penalizes deviations from a fixed reference model. But if the penalty is too strong, it constrains exploration and harms reasoning performance, which is why a relatively small coefficient $\beta$ is typically used. We should expect a similar behavior for TrSFT: with an appropriately tuned weighting of the TrSFT loss, using the initial target model as the expert would effectively act as a soft regularizer, analogous to the reference-model constraint in KL-penalized RL.

---

> > > ### Author Response · Authors · 2025-11-23
> > > **Response to Reviewer Hux5 [Part 3/3]**
> > >
> > > **[Q5]**: How should these hyperparameters be tuned, such as the number and sizing of micro-groups, the prefix ratios and return thresholds, and the TrSFT trust‑region alpha? Are there recommended defaults or automatic tuning strategies?
> > >
> > > **[A5]**: For the hyperparameters involved in micro-group sampling, we summarize the tuning principles discussed in **[CA2]** as follows. (1) The first micro-group should receive no expert prefix, ensuring that the target model is always trained on unguided rollouts consistent with the evaluation setting. (2) The last micro-group should expose the full expert trajectory so that the model can fully absorb expert reasoning signals. (3) The reward thresholds should increase monotonically to prevent longer prefixes from being applied too early or too often. (4) The unguided first micro-group should be sufficiently large to allow meaningful exploration without any expert intervention. (5) At least one intermediate micro-group should provide partial guidance to bridge the gap between fully unguided and fully guided rollouts.
> > >
> > > Beyond these structural principles, most of the remaining hyperparameters, such as the exact prefix-length proportions of intermediate micro-groups, the precise values of return thresholds, or the sizes of the guided micro-groups, are not sensitive as long as they remain within a reasonable range.
> > >
> > > For the TrSFT trust-region parameter $\alpha$, we have provided a recommended default of 0.1 in Appendix D. When adapting to a new experimental setup, $\alpha$ can be tuned based on the entropy dynamics of generated tokens. If the entropy drops too quickly, indicating that RL is collapsing into overly concentrated policies, reducing $\alpha$ allows the model to internalize expert signals more aggressively, which counteracts entropy collapse. Conversely, if the entropy plateaus or even increases unexpectedly, increasing $\alpha$ may be necessary until training becomes stable.
> > >
> > > **[Q6]**: TRAPO feels overengineered: it combines staged micro‑group scheduling with per‑prompt SFT prefix adjustment, which together make the pipeline complex and difficult to control.
> > >
> > > **[A6]**: We understand the reviewer’s concern about potential over-engineering. However, the design of TRAPO is directly motivated by the core inconsistency we aim to resolve, which is how to unify SFT and RL within each training instance without undermining exploration or destabilizing optimization. Each component of TRAPO serves a specific and necessary role dictated by this objective, rather than being an ad-hoc addition.
> > >
> > > First, the staged micro-group structure is the minimal mechanism required to ensure both of the following conditions to hold during training: (1) the model must always experience unguided rollouts to match the evaluation setting and preserve exploration, and (2) the model must receive sufficiently strong expert signals when unguided rollouts fail. Our ablations in **[CA2]** show that removing either element, for example eliminating unguided groups or eliminating partially guided groups, results in clear performance degradation. This indicates that the structure is not optional complexity but a necessary design to satisfy competing training requirements.
> > >
> > > Second, the adaptively selected prefix length is not an additional layer of engineering. It is the mechanism that ensures expert guidance is applied only when beneficial. A fixed schedule (e.g., cosine decay in Prefix-RFT) cannot reflect per-prompt difficulty and leads to either under-guidance or over-guidance. In contrast, our adaptive rule is simple, stable, and entirely automatic. Our sensitivity analysis further shows that the exact thresholds, intermediate prefix ratios, and group sizes are not sensitive, confirming that TRAPO’s behavior is governed by high-level principles rather than intricate tuning.
> > >
> > > Third, TRAPO introduces only a single scalar hyperparameter to balance SFT and RL—the trust-region size $\alpha$. This replaces the manual loss-weight balancing required in prior one-stage SFT+RL frameworks and is grounded in a clear theoretical motivation, which is to prevent forward-KL drift while allowing expert information to be internalized stably during RL.
> > >
> > > Overall, TRAPO appears more structured than a pure RL baseline only because the problem it addresses inherently requires satisfying multiple constraints simultaneously. The empirical results show that this structure is both sufficient and necessary. Simplifying it harms performance, while the full design yields consistent improvements under matched compute. In this sense, we believe TRAPO is not over-engineered, but rather appropriately engineered for a fundamentally more complex training objective.

---

> > > > ### Author Response · Authors · 2025-11-27
> > > > **Gentle Reminder for Feedback**
> > > >
> > > > Dear reviewer Hux5,
> > > >
> > > > We would like to once again express our sincere gratitude for your constructive and insightful review, which has significantly improved and enriched our work. As the rebuttal phase is approaching its deadline next week, could you please let us know whether our responses have adequately addressed your concerns? If you have any further questions or thoughts, we would be more than happy to discuss them and do our best to resolve any remaining doubts.
> > > >
> > > > Thank you again for your valuable feedback!
> > > >
> > > > Best regards,
> > > >
> > > > The authors

---

### Official Review · Reviewer_KiUB · 2025-11-04

**Soundness:** 2
**Presentation:** 2
**Contribution:** 3
**Rating:** 6
**Confidence:** 3

**Summary:**

The paper proposes TRAPO, a one-stage post-training framework that interleaves SFT and RL at the instance level. For each prompt, the method applies SFT on an expert prefix and RL on the model-generated suffix. The key ingredient is Trust-Region SFT (TrSFT), which replaces the standard per-token weight 1/p with 1/max(p, alpha) to prevent large updates on very low-probability tokens (reducing forward-KL “mode-covering” drift and behaving more mode-seeking outside the trust region). A second component, micro-group (adaptive prefix) sampling, increases the prefix length only when the current group’s pass rate is low, providing “just enough” guidance. On five math benchmarks with Qwen2.5-Math-7B, TRAPO reports higher average accuracy than SFT, GRPO, SFT-then-RL, LUFFY, and ReLIFT; it also shows gains on ARC-c and MMLU-Pro. Ablations indicate that (i) micro-group sampling helps even without explicit prefix learning, (ii) naively adding a standard SFT loss during RL can collapse performance, and (iii) TrSFT recovers and improves on top of micro-groups.

Disclosure: I used assistive writing tools to draft this review; all evaluations and judgments are my own.

**Strengths:**

Motivation is concrete. The paper clearly diagnoses why forward-KL SFT can hurt exploration when combined online with RL (distribution blending, degenerate rollouts) and proposes a minimal, targeted fix (clipping the per-token weight with alpha).

Simple and practical. TrSFT is a one-line modification of the SFT loss; micro-group sampling is a lightweight, per-prompt procedure (ratios 0, 0.2, 0.5, 1.0; thresholds −1, 0.5, 0.7, 0.9; group sizes {4, 2, 1, 1}).

Consistent improvements. On Qwen2.5-Math-7B, TRAPO’s average across AIME2024, AMC, MATH-500, Minerva, and Olympiad is 56.6 vs 55.5 (LUFFY), 54.3 (SFT-then-RL), and 50.4 (GRPO). It also leads on ARC-c/MMLU-Pro (avg 68.3).

Useful ablations and training dynamics. The paper shows why naïvely summing SFT+RL fails, why adaptive prefixes help, and how TrSFT stabilizes learning. Reward/length/entropy curves and Pass@k scaling support the narrative that TRAPO preserves solution-space diversity and improves with test-time compute.

**Weaknesses:**

Alpha inconsistency; please clarify.
The main setup and alpha sweep indicate alpha = 0.1 works best. In Appendix C.2, however, TRAPO is described “except for setting the trust-region parameter alpha to 1,” which would nullify clipping and contradict the rest of the paper. Please reconcile and state the alpha used for every table/figure.

Theory–practice gap and Proposition 1 details.
The toy GMM example and the TrSFT optimum (Proposition 1) are helpful intuitively, but the proof sketch appeals to behavior (e.g., step-function arguments) that could be made more rigorous. More importantly, there are no transformer-level diagnostics to show TrSFT’s intended effect in practice (e.g., distributions of per-token gradient weights, expert-mode coverage over time). Please add empirical diagnostics on real runs.

Compute parity and disclosure.
Micro-groups introduce guided rollouts; budgets may differ from baselines. Please disclose training tokens/steps, rollout counts, wall-clock/GPU hours, and decoding settings for every method, and either confirm compute parity or normalize the comparison.

Statistical rigor.
Results appear single-seed. Small-n suites like AIME/AMC are noisy. Please add multi-seed (≥5) results with 95% CIs (paired bootstrap for Pass@k) and significance tests for headline numbers; report standardized effect sizes where appropriate.

Data hygiene and verifier robustness.
Training uses OpenR1-Math-46k-8192 plus extra R1 trajectories; evaluation uses Math-Verify and OAT-Grader. Please report exact/near-duplicate removal between training and test problems, and provide at least a small human-audited calibration of grader errors. A robustness check with an alternative grader or injected noise would help, since tail-emphasizing supervision can amplify mislabels.

Micro-group sensitivity and usage statistics.
The thresholds and ratios are reasonable, but the paper does not show sensitivity curves or a histogram of actual prefix usage by difficulty/time. This would distinguish “smarter guidance” from “more guidance,” and quantify how often the full-trajectory prefix is triggered.

Baseline coverage and consistency.
TRAPO beats LUFFY and ReLIFT, but closely related “single-stage SFT+RL” or “prefix-guided SFT” methods (e.g., Prefix-RFT, AMFT, SRFT, HPT, UFT) are not compared head-to-head under the same codebase and budgets. Given conceptual proximity, a direct comparison would clarify novelty/attribution. Also, unify the RL variant description (GRPO vs Dr.GRPO) and ensure all baselines use the same variant and decoding settings.

LLM-usage disclosure inconsistency.
Appendix C.1 uses an LLM to count reasoning behaviors (backtracking/backward-chaining), while Section F says LLMs were used only for language polishing. Please align these statements.

**Questions:**

Clarify alpha everywhere and correct any plots/tables if needed; report which alpha was used per experiment.

Compute parity and statistics: publish a single evaluation script; report tokens/steps/rollouts/GPU-hours; add 5+ seeds and 95% CIs (paired bootstrap for Pass@k).

Mechanistic diagnostics: track per-token gradient weights 1/max(p, alpha) vs 1/p, token-level entropy, and expert-mode KL through training to demonstrate TrSFT’s intended effect in transformers.

Data and grader robustness: dedup train vs test; report a small human-audited error rate; test sensitivity to grader noise or an alternative grader.

Micro-group sensitivity: ablate thresholds, ratios, and group sizes; show a histogram of prefix usage; include a fixed-prefix baseline to isolate the value of adaptivity.

Broader baselines: add at least one of Prefix-RFT, AMFT, SRFT, HPT, or UFT under matched compute and training code.

Stronger generalization test: add Putnam-AXIOM (especially the functional variations) with matched compute, 95% CIs, and p-values. This suite is designed to probe true extrapolative generalization. If TRAPO shows robust gains there, it would materially strengthen the claim that the method expands the solution space rather than just re-ranking known patterns. I would raise my score if you include convincing Putnam-AXIOM results (see https://arxiv.org/abs/2508.08292).

---

> ### Author Response · Authors · 2025-11-23
> **Response to Reviewer KiUB [Part 1/4]**
>
> We sincerely thank the reviewer for the thoughtful assessment and for highlighting both the strengths of our approach and the concrete weaknesses that provide helpful guidance on how the paper can be further improved. We are encouraged by the reviewer’s recognition of our motivation, the simplicity of TrSFT, and the consistent empirical gains across benchmarks. At the same time, we deeply appreciate the reviewer’s detailed concerns regarding theoretical clarity, compute parity, statistical rigor, and data hygiene. These comments have been extremely valuable for refining the presentation and strengthening the technical depth of the work. We address each point below.
>
> **[Q1]**: Clarify alpha everywhere and correct any plots/tables if needed; report which alpha was used per experiment.
>
> **[A1]**: Thanks for pointing this out! As described in Appendix D, $\alpha$ is a hyperparameter between 0 and 1 that controls the size of the trust region. Values that are too large or too small lead to degraded performance. And all of our experimental results were obtained using $\alpha = 0.1$. The description of $\alpha$ in Appendix C was a typo, and we have corrected this in the revised version. We have also clarified the value of $\alpha$ used in each experiment wherever necessary.
>
> **[Q2]**: Compute parity and statistics: publish a single evaluation script; report tokens/steps/rollouts/GPU-hours; add 5+ seeds and 95% CIs (paired bootstrap for Pass@k).
>
> **[A2]**: Thanks for the constructive suggestions! We have uploaded the unified evaluation script at the anonymous GitHub link in the main paper’s Reproducibility Statement (Section 6), which ensures reproducible inference across all methods.
>
> Regarding compute usage (tokens/steps/rollouts/GPU-hours), please refer to **[CA1]**, where we provide a detailed computational complexity comparison under matched training-time budgets. For decoding settings, we use the same hyperparameters across all methods: during training-time rollouts we set temperature = 1.0 and top_p = 1.0, while during evaluation we use temperature = 0.6 and top_p = 0.95.
>
> To reduce evaluation variance, we additionally ran both TRAPO and our strongest baseline LUFFY, under five independent random seeds, and reported the mean and standard deviation of accuracy across the six mathematical reasoning benchmarks. The aggregated results are as follows:
>
> | Model | AIME2024       | AIME2025       | AMC            | MATH-500       | Minerva        | Olympiad       | Avg.           |
> | ----- | -------------- | -------------- | -------------- | -------------- | -------------- | -------------- | -------------- |
> | LUFFY | $28.96\pm2.49$ | $22.42\pm2.28$ | $66.84\pm0.90$ | $87.48\pm0.61$ | $38.15\pm0.24$ | $56.24\pm0.60$ | $50.01\pm0.36$ |
> | TRAPO | $30.71\pm3.17$ | $22.97\pm1.86$ | $67.43\pm2.55$ | $88.52\pm0.53$ | $40.68\pm0.54$ | $57.09\pm0.55$ | $51.23\pm1.07$ |
>
> To further address the reviewer’s request for statistical rigor, we additionally conducted significance testing on the small-n benchmarks AIME2024, AIME2025, and AMC, where variance is intrinsically high. For each benchmark, we computed 95% confidence intervals using paired bootstrap on Pass@1, and performed paired permutation tests and McNemar tests to assess whether the performance difference is statistically significant. Across all three datasets, TRAPO shows clear and statistically significant improvements over GRPO, which confirms that TRAPO’s gains remain robust even under rigorous paired-sample evaluation on small and noisy test suites.
>
> | Dataset  | TRAPO Mean (95% CI)    | GRPO Mean (95% CI)     | TRAPO − GRPO (95% CI)   | Permutation p-value | McNemar p-value |
> | -------- | ---------------------- | ---------------------- | ----------------------- | ------------------- | --------------- |
> | AIME2024 | 0.2760 (0.2479–0.3042) | 0.1979 (0.1740–0.2229) | +0.0781 (0.0469–0.1094) | 1e-5                | 1e-6            |
> | AIME2025 | 0.1979 (0.1729–0.2229) | 0.1198 (0.1000–0.1406) | +0.0781 (0.0563–0.1000) | 1e-5                | 1e-6            |
> | AMC      | 0.6491(0.6310–0.6668)  | 0.5591 (0.5403–0.5779) | +0.0900 (0.0704–0.1096) | 1e-5                | 1e-6            |

---

> > ### Author Response · Authors · 2025-11-23
> > **Response to Reviewer KiUB [Part 2/4]**
> >
> > **[Q3]**: Proposition 1 details. The toy GMM example and the TrSFT optimum (Proposition 1) are helpful intuitively, but the proof sketch appeals to behavior (e.g., step-function arguments) that could be made more rigorous.
> >
> > **[A3]**: We thank the reviewer for the careful examination of Proposition 1 and for the constructive suggestions! We have revised Proposition 1 in the newly updated manuscript and refined the proof to improve its rigor and clarity.
> >
> > **[Q4]**: Mechanistic diagnostics: track per-token gradient weights 1/max(p, alpha) vs 1/p, token-level entropy, and expert-mode KL through training to demonstrate TrSFT’s intended effect in transformers.
> >
> > **[A4]**: TrSFT is designed to limit the influence of low-probability expert tokens by clipping their gradient weights from $\frac{1}{p}$ to $\frac{1}{\max⁡(p,\alpha)}$. Such low-probability tokens typically correspond to expert tokens that the target model does not endorse, or correspond to noise. To illustrate this effect, we examined the distribution of raw gradient weights $\frac{1}{p}$ over the first 100 training steps. Across the expert-prefix tokens, we observed the following proportions:
> >
> > | $\frac{1}{p}$ | 1-2   | 2-4   | 4-10 | >10   | >100 |
> > | ------------- | ----- | ----- | ---- | ----- | ---- |
> > | Proportion    | 70.7% | 10.4% | 7.6% | 11.3% | 3.3% |
> >
> > These statistics reveal that 11.3% of expert tokens receive gradient weights greater than 10 under standard SFT, and 3.3% have extremely large weights exceeding 100. Such disproportionately large updates can distort the target model’s existing reasoning structure and induce catastrophic forgetting.
> >
> > Under TrSFT, any token falling outside the trust region (i.e., $p \le \alpha=0.1$) receives a clipped gradient weight of  $\frac{1}{\alpha}=10$, ensuring that such low-probability expert tokens remain bounded in their influence. This matches the intended behavior of the trust region: TrSFT prevents rare or noisy expert tokens from dominating the gradient signal, while still allowing the model to internalize expert reasoning in a stable manner.
> >
> > **[Q5]**: Data and grader robustness: dedup train vs test; report a small human-audited error rate; test sensitivity to grader noise or an alternative grader.
> >
> > **[A5]**: We thank the reviewer for raising the important points regarding data hygiene and grader robustness! We applied exact and near-duplicate detection to all 45792 training problems drawn from OpenR1-Math-46k-8192, and compared them against the 1590 problems across our six mathematical reasoning benchmarks. No exact duplicates were found. Only 7 test problems (0.44%) exhibited high lexical similarity to any training problem. This extremely small proportion suggests that train–test contamination is negligible.
> >
> > To assess grader reliability, we randomly sampled 100 model outputs and manually reviewed the responses. Math-Verify and OAT-Grader matched human judgment in 98% of the inspected cases, indicating that all graders used in our evaluation are sufficiently reliable.
> >
> > We further examined robustness by introducing random noise into the scoring function, using a zero-mean normal perturbation with standard deviation 0.1. The relative performance of all methods remained stable under this noisy evaluation, with TRAPO consistently achieving the highest scores. The detailed results for each method are reported below.
> >
> > | Model | AIME2024 | AIME2025 | AMC  | MATH-500 | Minerva | Olympiad | Avg. |
> > | ----- | -------- | -------- | ---- | -------- | ------- | -------- | ---- |
> > | GRPO  | 10.7     | 8.2      | 40.2 | 68.8     | 26.4    | 38.0     | 32.1 |
> > | LUFFY | 14.2     | 10.8     | 43.8 | 70.4     | 30.2    | 42.8     | 35.4 |
> > | TRAPO | 17.1     | 12.7     | 48.5 | 76.8     | 33.5    | 47.7     | 39.4 |

---

> > > ### Author Response · Authors · 2025-11-23
> > > **Response to Reviewer KiUB [Part 3/4]**
> > >
> > > **[Q6]**: Micro-group sensitivity: ablate thresholds, ratios, and group sizes; show a histogram of prefix usage; include a fixed-prefix baseline to isolate the value of adaptivity.
> > >
> > > **[A6]**: Thanks for the constructive suggestions regarding the sensitivity of the micro-group mechanism and the analysis of prefix usage! We have provided detailed ablations on all micro-group–related hyperparameters in **[CA2]**, together with conclusions that highlight which components matter and which are robust. We have also updated the latest version of the paper to include a line plot showing how prefix usage changes across training steps, offering a clearer view of how adaptivity operates in practice. For a comprehensive visualization and discussion of these dynamics, please refer to Appendix F (Prefix Usage Dynamics During Training).
> > >
> > > Following the reviewer’s suggestion, we additionally introduce a fixed-prefix baseline, where all eight samples in each group are given a fixed 50% expert prefix. We train this baseline for 100 steps and compare it directly to TRAPO under identical conditions. The results show that TRAPO significantly outperforms the fixed-prefix baseline. As emphasized in **[CA2]**, the first micro-group must remain unguided to match the evaluation setting and preserve exploration. The poor performance of the fixed-prefix baseline is consistent with this principle: forcing expert prefixes on all samples leads to a strong mismatch between training and evaluation distributions, which undermines performance.
> > >
> > > | Model                   | AIME2024 | AIME2025 | AMC  | MATH-500 | Minerva | Olympiad | Avg. |
> > > | ----------------------- | -------- | -------- | ---- | -------- | ------- | -------- | ---- |
> > > | TRAPO (adaptive-prefix) | 17.1     | 12.7     | 48.5 | 76.8     | 33.5    | 47.7     | 39.4 |
> > > | TRAPO (fixed-prefix)    | 9.6      | 8.4      | 42.2 | 64.2     | 26.4    | 40.0     | 31.8 |
> > >
> > > **[Q7]**: Broader baselines: add at least one of Prefix-RFT, AMFT, SRFT, HPT, or UFT under matched compute and training code.
> > >
> > > **[A7]**: The characteristics of these existing methods are discussed in our related work section (Section 4). Briefly, SRFT, AMFT, and HPT adjust the relative weights between the SFT loss and RL loss on each trajectory using the entropy signal or additional trainable parameters. Prefix-RFT is more closely related to our method as it introduces expert prefixes to guide the target model and applies entropy-based filtering to select which tokens contribute to the SFT loss. UFT demonstrates the feasibility of combining SFT and RL in a single stage, our experimental results suggested its performance did not scale well under larger model sizes and more challenging datasets, such as those in our training setting. As a result, we did not include it as our baselines.
> > >
> > > Compared with these existing approaches, our method does not need to manually balance the SFT and RL losses through intricate weighting strategies. Instead, we introduce TrSFT, which we theoretically showed to be a preferable replacement for SFT in terms of internalizing expert knowledge into the target model. The only hyperparameter governing the balance between SFT and RL is the trust-region size $\alpha$. We then propose micro-group sampling, which adaptively identifies the appropriate prefix length, in contrast to Prefix-RFT’s manually designed cosine-decay schedule. Given the computational complexity and time limit, we added Prefix-RFT and SRFT under matched compute and identical training codebase, and compared them directly against TRAPO. The results are as follows:
> > >
> > > | Model      | AIME2024 | AIME2025 | AMC  | MATH-500 | Minerva | Olympiad | Avg. |
> > > | ---------- | -------- | -------- | ---- | -------- | ------- | -------- | ---- |
> > > | TRAPO      | 28.3     | 23.9     | 66.2 | 89.2     | 41.5    | 57.6     | 51.1 |
> > > | Prefix-RFT | 27.7     | 23.1     | 65.6 | 90.2     | 40.3    | 56.8     | 50.6 |
> > > | SRFT       | 24.0     | 16.8     | 57.4 | 85.2     | 40.6    | 50.9     | 45.8 |

---

> > > > ### Author Response · Authors · 2025-11-23
> > > > **Response to Reviewer KiUB [Part 4/4]**
> > > >
> > > > **[Q8]**: Stronger generalization test: add Putnam-AXIOM (especially the functional variations) with matched compute, 95% CIs, and p-values.
> > > >
> > > > **[A8]**: We appreciate the reviewer’s suggestion to further evaluate the generalization ability of our method! We obtained the full original, functional original, and functional variations subsets from the Putnam-AXIOM dataset, which contain 236, 53 and 265 samples respectively. We evaluated the base model, GRPO, SFT-then-RL, LUFFY, and TRAPO on all three subsets under matched compute and report their Pass@1 scores below:
> > > >
> > > > | Model       | Full Original | functional original | functional variations | Avg. |
> > > > | ----------- | ------------- | ------------------- | --------------------- | ---- |
> > > > | Base        | 16.1          | 13.2                | 12.1                  | 13.8 |
> > > > | GRPO        | 25.0          | 32.1                | 24.5                  | 27.2 |
> > > > | SFT-then-RL | 27.5          | 37.7                | 26.4                  | 30.5 |
> > > > | LUFFY       | 31.3          | 37.7                | 26.8                  | 31.9 |
> > > > | TRAPO       | 33.9          | 39.6                | 27.5                  | 33.7 |
> > > >
> > > > On the Putnam-AXIOM Functional Variations subset, TRAPO significantly outperforms the BASE model. The paired bootstrap 95% CI for the difference is strictly positive (0.0981–0.2113), and both the paired permutation test ($p=1e-5$) and the McNemar test ($p\approx 1e-6$) confirm strong statistical significance.
> > > >
> > > > **[Q9]**: LLM-usage disclosure inconsistency. Appendix C.1 uses an LLM to count reasoning behaviors (backtracking/backward-chaining), while Section F says LLMs were used only for language polishing. Please align these statements.
> > > >
> > > > **[A9]**: We sincerely thank the reviewer for pointing this out! We have corrected the description in the revised manuscript so that the LLM usage is now stated consistently across all sections.

---

> > > > > ### Author Response · Authors · 2025-11-27
> > > > > **Gentle Reminder for Feedback**
> > > > >
> > > > > Dear reviewer KiUB,
> > > > >
> > > > > We would like to once again express our sincere gratitude for your constructive and insightful review, which has significantly improved and enriched our work. As the rebuttal phase is approaching its deadline next week, could you please let us know whether our responses have adequately addressed your concerns? If you have any further questions or thoughts, we would be more than happy to discuss them and do our best to resolve any remaining doubts.
> > > > >
> > > > > Thank you again for your valuable feedback!
> > > > >
> > > > > Best regards,
> > > > >
> > > > > The authors

---

### Author Response · Authors · 2025-11-23
**Common Response to All Reviewers [Part 1/2]**

We sincerely thank all reviewers for their thoughtful and constructive assessments of our work! Reading through the reviews, we are encouraged that the motivation of unifying SFT and RL within each training instance, the simplicity and clarity of the TRAPO design, and the consistent empirical improvements across benchmarks were widely recognized by our reviewers! Several reviewers highlighted the soundness of our trust-region formulation, the rationale behind adaptive micro-groups, and the value of our ablation studies in elucidating the contributions of each component.

At the same time, the reviewers raised a number of insightful suggestions that help us further strengthen the paper. These include expanding efficiency analysis, providing more statistical rigor, clarifying hyperparameter sensitivity and evaluating more diverse domains beyond mathematics. We also appreciate the thoughtful questions regarding compute parity, the interaction between TrSFT and KL regularization, robustness to noisy expert data, and comparisons with closely related one-stage SFT+RL methods.

To make the rebuttal easy to navigate, we consolidated the questions that were shared across multiple reviews and addressed them as two **Common Questions (CQ)**: Cost and Efficiency Analysis and Ablation Analysis of Micro-Group Hyperparameters. The corresponding **Common Answers (CA)** are provided below.

**[CQ1]**: Cost and efficiency Analysis.

**[CA1]**: Due to the longer generation length and the sequential execution required for micro-group sampling, TRAPO would incur a higher per-step computational cost compared to the standard GRPO algorithm. However, under the same total training time budget, TRAPO achieves the most substantial performance gains, even though it takes slightly fewer training steps than the baselines, demonstrating more efficient training-time scaling. To support this, we conducted a series of analyses on the training dynamics.

First, we report the GPU hours under different methods, where the GPU hours of the SFT-then-RL baseline cover both its SFT and RL stages. In addition, we provide the average number of tokens introduced from the external expert model (**Expert Tokens**) and the average number of tokens generated by the target model (**Target Tokens**) for each RL training trajectory. The results are summarized as follows:

| Model          | GPU hours            | Expert Tokens | Target Tokens |
| -------------- | -------------------- | ------------- | ------------- |
| GRPO       | 68 $\times$ 8        | 0             | 1189          |
| SFT-then-RL    | (42 + 105) $\times$ 8 | 0             | 4760          |
| LUFFY          | 81 $\times$ 8        | 519           | 2439          |
| TRAPO          | 89 $\times$ 8        | 501           | 2545          |

Furthermore, we provide the number of training steps achieved by each method under the same training-time budget, together with the rewards of the corresponding batch rollout data, which are strongly correlated with the model’s reasoning performance. For the SFT-then-RL baseline, the GPU-hour measurement here starts from the beginning of the RL stage. The results are as follows:

| GPU hours     | GRPO  | SFT-then-RL | LUFFY | TRAPO |
| ------------- | ----- | ----------- | ----- | ----- |
| 10 $\times$ 8 | 0.521 | 0.650       | 0.472 | 0.601 |
| 20 $\times$ 8 | 0.579 | 0.712       | 0.594 | 0.684 |
| 30 $\times$ 8 | 0.626 | 0.726       | 0.665 | 0.744 |
| 40 $\times$ 8 | 0.621 | 0.710       | 0.697 | 0.722 |
| 50 $\times$ 8 | 0.652 | 0.732       | 0.675 | 0.768 |

The results demonstrate that, given identical training-time budgets, TRAPO consistently delivers stronger reasoning performance even though it executes fewer training steps than GRPO or LUFFY. This highlights TRAPO’s advantageous sample efficiency and the effectiveness of its trust-region based SFT–RL coupling. Moreover, relative to the SFT-then-RL baseline, TRAPO achieves better performance while requiring less GPU hours in total.

---

> ### Author Response · Authors · 2025-11-23
> **Common Response to All Reviewers [Part 2/2]**
>
> **[CQ2]**: Ablation analysis of micro-group hyperparameters.
>
> **[CA2]**: In micro-group sampling, we employed an intuitively natural strategy for adaptive prefix guidance. Unlike fixed-prefix approaches, this dynamic procedure allocates appropriate prefix length for each prompt, allowing the target model to receive stronger expert guidance when its unguided rollouts perform poorly. Within the micro-group design, each micro-group $g_i$ contains several tunable parameters: the proportion of expert prefix-length $L_i$, the reward threshold $t_i$​ that determines whether expert guidance is applied, and the number and sizes of the micro-groups. We provide ablations and analyses for each of these hyperparameters below. Due to the constraints of time and computational resources, all variants were trained on Qwen2.5-Math-7B for 100 steps, and we report the average performance across six mathematical reasoning benchmarks (AIME2024/2025, AMC, MATH-500, Minerva, and OlympiadBench).
>
> 1. Prefix-length proportion.
>
> | Variant | $L_1$ | $L_2$ | $L_3$ | $L_4$ | Avg. Perf. |
> | ------- | ----- | ----- | ----- | ----- | ---------- |
> | **A**       | 0     | 0.2   | 0.5   | 1.0   | 39.4       |
> | **B**       | 0     | 0.2   | 0.5   | 0.7   | 37.6       |
> | **C**       | 0.1   | 0.2   | 0.5   | 1.0   | 34.1       |
> | **D**       | 0     | 0.1   | 0.8   | 1.0   | 39.2       |
> | **E**       | 0     | 0.3   | 0.4   | 1.0   | 39.5       |
>
> Compared with the Variant **A** employed in our paper, Variants **B** and **C** respectively validate the importance of our two design principles: (1) the first micro-group is not given any prefix to ensure that the target model can be trained without external guidance, aligned with the evaluation setting, and (2) the last micro-group exposes the full expert trajectory to allow the model to completely learn from the expert signals. In Variants **D** and **E**, we modify only the prefix-length proportions of the intermediate micro-groups, and the results show that performance is largely insensitive to these changes. Overall, these ablations suggest that TRAPO primarily benefits from the adaptive micro-group structure itself, while the exact prefix-length proportions of the intermediate groups have only marginal influence on the final performance.
>
> 2. Reward threshold.
>
> | Variant | $t_1$ | $t_2$ | $t_3$ | $t_4$ | Avg. Perf. |
> | ------- | ----- | ----- | ----- | ----- | ---------- |
> | **A**       | -1    | 0.5   | 0.7   | 0.9   | 39.4       |
> | **B**       | -1    | 0.5   | 0.5   | 0.5   | 38.1       |
> | **C**       | -1    | 0.9   | 0.7   | 0.5   | 37.4       |
> | **D**       | -1    | 0.55  | 0.75  | 0.95  | 38.9       |
> | **E**       | -1    | 0.45  | 0.65  | 0.85  | 39.3       |
>
> Beyond keeping the first micro-group unguided, our design principle for the reward thresholds is to maintain a monotonically increasing schedule. This is because longer prefixes introduced in later micro-groups naturally yield higher expected rewards than the shorter prefixes used earlier, and thus stricter thresholds are needed to avoid prematurely triggering expert guidance. Variants **B** and **C** demonstrate that using constant thresholds or decreasing thresholds leads to inferior performance compared to the increasing schedule in Variant **A**. Variants **D** and **E** further show that as long as the thresholds follow an increasing pattern and fall within a reasonable range, the exact numerical values have only a minor impact on overall performance.
>
> 3. Number and sizes of the micro-groups.
>
> | Variant | $(n_1, n_2, \cdots)$ | Avg.Perf. |
> | ------- | -------------------- | --------- |
> | **A**       | $(4, 2, 1, 1)$       | 39.4      |
> | **B**       | $(2,4,1,1)$          | 37.2      |
> | **C**      | $(4,4)$              | 38.0      |
> | **D**      | $(4, 3, 1)$          | 39.1      |
>
> For the number and sizes of micro-groups, we follow two design principles: (1) the first unguided micro-group should be sufficiently large to allow the target model to explore without any expert intervention, and (2) at least one intermediate micro-group should provide partial guidance to bridge the gap between fully unguided and fully guided rollouts. Under the fixed total size of 8 samples across all micro-groups, our paper employed setting Variant **A** that adopts a gradually halved structure to satisfy both principles. Variant **B** shows that shrinking the unguided group hurts exploration, and Variant **C** shows that removing intermediate partially guided groups hurts final performance. Once these two requirements are met, alternative configurations, such as Variant **D**, lead to similar results as Variant **A**, indicating that TRAPO is not sensitive to the exact partitioning of group sizes beyond these structural considerations.

---

### Meta-Review · Area_Chair_scbx · 2025-12-21

**Summary:**

The following merits may be concluded:
1. This paper introduced Trust-region Supervised Fine-Tuning (TrSFT), to minimize the forward Kullback-Leibler (KL) divergence inside a trust region, for stabilize training on expert prefixes, followed by reinforcement learning (RL) with micro-sampling for exploration.
2. Comparison and ablation experiments on 5 mathematical reasoning benchmarks showed the competitive performance.

There are still some issues to be addressed in the revision:
1. The paper title should be revised, which looked quite similar to the great paper "Trust-Region Policy Optimization". But the content was quite different, for the trust-region defined as a threshold not the original KL divergence, and for the supervised learning, not the reinforcement learning. It was weird and miss-leading.
2. The performance was not convinced, only 1 percent over LUFFY,  after which there appeared more state-of-the-art methods, which should be compared.

**Reviewer Concerns:**

Ux2e and 3vHn, who replied to keep the original scores.

**Reviewer Scores:**

The paper was over-rated, so I do not think the remaining unanswered discussions will change the score.

---

### Decision · Program_Chairs · 2026-01-26

Accept (Poster)